# Bronchial epithelial transcriptomics and experimental validation reveal asthma severity-related neutrophilc signatures and potential treatments

Qian Yan[1,2,3,4,6], Xinxin Zhang[1,2,3,4,6], Yi Xie[1,2], Jing Yang[1,2,3,4], Chengxin Liu[1,2], Miaofen Zhang[1,2,3,4], Wenjiang Zheng[1,2], Xueying Lin[1,2], Hui-ting Huang[1], Xiaohong Liu[1], Yong Jiang [5✉], Shao-feng Zhan [1✉] & Xiufang Huang [1,2,3,4✉]

Airway epithelial transcriptome analysis of asthma patients with different severity was used to disentangle the immune infiltration mechanisms affecting asthma exacerbation, which may be advantageous to asthma treatment. Here we introduce various bioinformatics methods and develop two models: an OVA/CFA-induced neutrophil asthma mouse model and an LPS-induced human bronchial epithelial cell damage model. Our objective is to investigate the molecular mechanisms, potential targets, and therapeutic strategies associated with asthma severity. Multiple bioinformatics methods identify meaningful differences in the degree of neutrophil infiltration in asthma patients with different severity. Then, PTPRC, TLR2, MMP9, FCGR3B, TYROBP, CXCR1, S100A12, FPR1, CCR1 and CXCR2 are identified as the hub genes. Furthermore, the mRNA expression of 10 hub genes is determined in vivo and in vitro models. Reperixin is identified as a pivotal drug targeting CXCR1, CXCR2 and MMP9. We further test the potential efficiency of Reperixin in 16HBE cells, and conclude that Reperixin can attenuate LPS-induced cellular damage and inhibit the expression of them. In this study, we successfully identify and validate several neutrophilic signatures and targets associated with asthma severity. Notably, Reperixin displays the ability to target CXCR1, CXCR2, and MMP9, suggesting its potential therapeutic value for managing deteriorating asthma.

[1] The First Affiliated Hospital of Guangzhou University of Chinese Medicine, Guangzhou, China. [2] The First Clinical Medical School of Guangzhou University of Chinese Medicine, Guangzhou, China. [3] Lingnan Medical Research Center of Guangzhou University of Chinese Medicine, Guangzhou, China. [4] Guangdong Provincial Clinical Research Academy of Chinese Medicine, Guangzhou, China. [5] Shenzhen Hospital of Integrated Traditional Chinese and Western Medicine, Shenzhen, China. [6] These authors contributed equally: Qian Yan, Xinxin Zhang. ✉email: jiangyongszzzy@163.com; zsfstone@163.com; huangxiufang@gzzyydx17.wecom.work

Asthma is a common chronic inflammatory airway disease and a major public health problem. The global age-standardized prevalence of asthma has remarkably risen by 3.6% over the past decade[1]. Even worse, the downward trend in asthma-related deaths has stagnated since the late 1980s, indicating that asthma remains a major threat to human health[2]. Asthma is a heterogeneous disease mainly characterized by chronic airway hyperresponsiveness and airway inflammation[3]. Major advances have been made in the treatment of asthma with inhaled corticosteroids[4]. However, the use of corticosteroids is often limited in severe asthma (SA) and asthma characterized by neutrophil infiltration[5]. Therefore, there is a great demand to elucidate the molecular mechanisms and discover new therapeutic targets and agents to help improve asthma treatment and management.

The airway epithelium serves as the first physical barrier between the human body and the external environment, releasing various cytokines and chemokines to initiate and regulate innate and adaptive immune processes in the host[6]. The airway epithelium responds to the external microbial and viral invasions by activating the immune system and plays a critical role in repairing the airway[7]. Given that airway epithelium plays an important immunomodulatory role in asthma, it is valuable to evaluate the differences in pivotal genes and immune cell infiltration in airway epithelium of patients with SA, mild-to-moderate asthma and healthy controls. Thus, we aimed to elucidate the molecular mechanisms and provide potential therapeutic approaches for asthma exacerbations.

A systems biology approach called weighted gene co-expression network analysis (WGCNA) can be used for identifying the critical gene modules of high relevance in microarray samples. More importantly, WGCNA can link modular genes to external sample characteristics, thus helping to capture remediation targets for SA[8]. The ssGSEA method can quantify the infiltration of 28 immunocytes in tissues by gene set variation analysis[9], and has become a commonly used technique in immunology. At the same time, we used various methods to evaluate the extent of immune cell infiltration in asthma of varying severity. At present, more and more studies are now focusing on the immune infiltration mechanisms of asthma. Leon A. Sokulsky et al. hold the view that targeting the *CXCL3/CXCL5/CXCR2* axis may provide a new therapeutic approach to alleviate rhinovirus-induced SA[10]. Doumet Georges Helou et al. suggest that leukocyte-associated immunoglobulin-like receptor 1, an immune checkpoint inhibitor, can activate type 2 innate lymphocytes and regulate airway hyperresponsiveness in type 2 asthma[11]. Therefore, we aimed to confirm new immune infiltration mechanisms, molecular biomarkers, pathways and potential drugs related to asthma severity based on the transcriptomics data from bronchial epithelium. Finally, OVA/CFA-induced asthma model dominated by neutrophils and human bronchial epithelial cell (16HBE) injury model induced by LPS were used to verify the hub genes and drugs for asthma, and the Fig. 1 displayed the course of our study.

## Methods

**Microarray data**. We selected gene expression datasets from the Gene Expression Omnibus (GEO, https://www.ncbi.nlm.nih.gov/geo/) database[12] of the National Center for Biotechnology Information. The criteria for the dataset were set as follows: (a) the dataset must contain healthy controls, mild, moderate and SA patients; (b) the samples in the dataset must be derived from airway epithelium. Since this study used only secondary data from public databases, it did not require approval from the Research Ethics Committee.

Transcriptomics data from 3 samples of airway epithelium brushings, sputum, and bronchoalveolar lavage (BAL) fluid were included in GSE89809[13] of the GPL13158 platform. According to our purpose, we excluded non-bronchial epithelium samples from the analysis. Thus, gene expression data from epithelial brushings including 18 healthy controls, 14 patients with mild asthma, 13 patients with moderate asthma and 11 patients with SA were finally chosen. The clinical information was shown in Supplementary Table 1. In addition, we obtained 1793 immune-related genes from the Immunology Database and Analysis Portal (ImmPort, https://www.immport.org/shared/home, updated: July 2020)[14].

**Immune cell infiltration analysis**. To clarify the distinction of immune cells between healthy individuals, mild, moderate asthma and SA, we used the ssGSEA[9], CIBERSORTx[15], Consensus$^{TME}$[16], MCP counter[17] and xCell[18] methods to evaluate the level of immune infiltration in each group in the GSE89809 dataset.

To further understand the immune infiltration mechanisms in asthma pathogenesis, an analysis of the Spearman correlation between identified genes and immune cell infiltration levels was carried out using the "p.adjust" R function. A P-value <0.05 was chosen as the filter criterion.

**Differential expression and enrichment analyses**. We conducted differential expression analysis using the "limma" package in R (version 4.3.0) to identify differentially expressed genes (DEGs) between the two distinct groups of individuals. DEGs were identified based on the following criteria: |Log$_2$ (fold change) FC| > 0.5 and P-value < 0.05. The differential expression of DEGs was visualized as volcano plots by the "ggplot2" package in R.

To identify the regulatory mechanisms associated with hub genes, we used the R package "clusterProfiler" to carry out enrichment analyses[19]. The screening criterion for Biological Process (BP) or Kyoto Encyclopedia of the Genome (KEGG) terms was an adjusted P-value of less than 0.05.

**WGCNA**. A weighted gene co-expression network was established by the "WGCNA" package in R[8]. First, missing values were confirmed by goodSampleGenes and free samples were determined by cluster analysis of 50 samples. The filtering criterion was R-square = 0.85. In order to obtain the maximum weighting coefficient $\beta$ and the perfect scale-free network, a soft threshold (power) plot was plotted as well as an average connectivity graph. We constructed the modules by cutting method of dynamic hybrid, and then clustered and merged similar modules. The parameters were set as follows: mergeCutHeight=0.25, verbose=3, minModuleSize=60 and deepSplit=2. Neutrophils were also included in the analysis and finally a Pearson correlative analysis was carried out between the clinical characteristics and modules.

**Protein-protein interaction (PPI) and GeneMANIA networks construction**. A PPI network was built to display the interaction between protein targets in the crucial modules by the STRING database (version 11.0, https://www.string-db.org/). We set the minimum required interaction score to 0.4 (medium confidence) and visualized them using the Cytoscape software (version 3.8.2). In addition, we used the GeneMANIA database[20] (https://genemania.org/) to construct gene-gene interaction (GGI) networks as well as co-expression and enrichment pathways interacting with hub genes. GeneMANIA database can further demonstrate the interactions between these hub targets, such as co-expression, co-localization, pathways, predicted physical

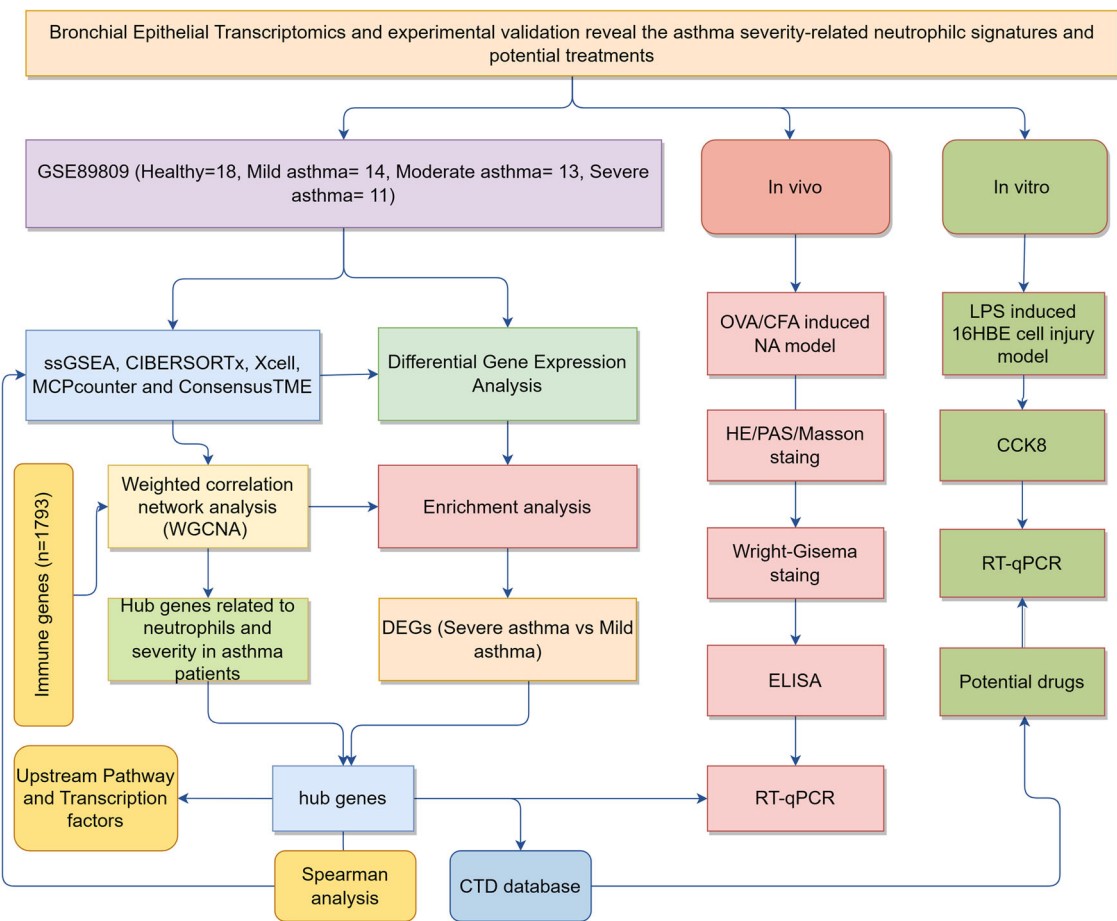

**Fig. 1 Flowchart of the systematic approach.** The procedure consisted of three steps: bioinformatics analysis and in vitro and in vivo experimental verifications. NA neutrophil asthma, DEG differential expression genes.

interactions, shared protein domains and genetic interactions. Each interaction was shown in a different color.

**Upstream pathway activity inference and transcription factor (TF)-gene network construction.** Signaling Pathway Enrichment using Expression Dataset version 2 (SPEED2, https://speed2.sys-bio.net/)[21] is a signal pathway enrichment analysis tool. SPEED2 is often used to quantify whether the input gene is enriched for strong pathway disorder signature genes to infer its upstream pathway activity. The Bates test was used to quantify the average rank change in the enrichment statistic.

The interaction network of TF-gene pairing was discovered by NetworkAnalyst (https://www.networkanalyst.ca/, version 3.0)[22]. The ENCODE database (https://www.encodeproject.org/) was used for the basic information about the TF-gene interaction network, which was further visualized by Cytoscape software (version 3.8.2)[23].

**Identification of predicted drug-target network.** ToppGene Suite (https://toppgene.cchmc.org)[24] is a one-stop portal for enrichment analysis and prioritization of input target genes based on functional annotation and PPI. The pharmacogenetic association data were derived from Connectivity Map (CAMP, https://portals.broadinstitute.org/cmap/)[25], Comparative Tox-icogenomics Database (CTD; http://CTD.mdibl.org)[26] and Drugbank (https://go.drugbank.com/)[27]. Then, we selected the CTD database for the next step in the analysis. Bonferroni-corrected hypergeometric distributions were used as the standard method to determine statistical significance.

**Establishment of a mouse model of Neutrophilic Asthma (NA).** The study was approved by the Laboratory Animal Ethics Committee of Guangzhou University of Chinese Medicine (Approval No. 20230413001). Female BALB/C mice aged 6-8 weeks were obtained from Zhejiang Weitong Lihua Laboratory Animal Technology Co., Ltd. (license number: SCXK (Zhejiang) 2019-0001, Guangzhou, China). The mice had ad libitum access to food and water throughout the day. The mice were randomly assigned to control or model groups. The ambient temperature ranged from 22 to 24 degrees Celsius, with a relative humidity of 40%-50%. Prior to the experiment, all mice were housed and acclimatized under standard conditions for 7 days, during which time they received distilled water and standard chow.

Based on the previously described models, we constructed and modified an OVA/CFA-induced NA mouse model[28] as follows: On the 0, 7 and 14 days, each mouse was immunized by intraperitoneal injection (i.p.) with 20 ug OVA (Grade III, Sigma, A5530, #9006-59-1) and 75 uL complete Freund's adjuvant (CFA, Sigma, #F5881). From day 21 to day 30, mice were placed in a mouse animal spray (HY-JSE01, 0.2 mL/min) for atomization (5% OVA, 40 min/day). Controls received the same amount of saline during the sensitization and stimulation stages. Mice were killed 24 h after the last OVA challenge.

**Histological staining.** The HE, PAS, and Masson staining techniques are widely used for histopathologic evaluation of the lungs. Specifically, these staining methods are used to assess the cellular morphology, mucus production and collagen deposition around the airways and blood vessels. The HE staining technique

provides information about overall tissue structure, including the presence of inflammatory cells such as neutrophils, lymphocytes and macrophages. PAS staining helps identify mucus-producing goblet cells and detect glycogen-rich structures. Masson staining helps to distinguish different types of collagen within the lung tissues, making it an effective tool for assessing lung fibrosis.

**BAL fluid cells collection and counting.** A transverse incision was made in the trachea and the whole lung BAL was obtained by rinsing 3 times consecutively with 0.6 mL of normal saline. The BAL was centrifuged at 1000 rpm for 10 min at 4 °C, and the resulting supernatant was collected and stored at −80 °C. The cell pellet was resuspended in 500 uL of erythrocyte lysate (biosharp, #BL503B) and centrifuged at 3000 rpm for 10 min. It was then washed with 500 uL of normal saline and centrifuged again under the same conditions. The BAL fluid was resuspended with 50 uL of normal saline, and cell slides were prepared using an automated smear centrifuge (CENCE, TXD3). BAL fluid samples were stained by the Wright-Giemsa method (Biosharp, #BL800A), and cell slides were scanned and photographed at 200× and 400× magnification using a Panoramic 250 Flash II Slide Scanner (3D HISTEC, Pannoramic MIDI, Budapest, Hungary). Total cell counts in the BAL fluid were determined using the Countstar system (BioLab).

**ELISA.** The levels of IgE (QuantiCyto, Cat#: EMC117), IL-8 (QuantiCyto, Cat#: EMC104.96), IL-1$\beta$ (QuantiCyto, Cat#: EMC001b.96), TNF-$\alpha$ (QuantiCyto, Cat#: EMC102a) and MPO (Multisciences, Cat#: EK2133S) in serum or BAL were detected by ELISA following the manufacturer's instructions.

**RT-qPCR.** Lung tissues were homogenized and total RNA was extracted with TrizolTM Reagent (Invitrogen, Cat#: 15596026). The mRNA levels were determined using the $2^{-\Delta\Delta CT}$ comparative method and normalized using $\beta$-actin as an internal control. The expression of target genes was verified by RT-qPCR. The primer sequences were provided in Supplementary Table 2.

**Cell culture and treatment.** Human bronchial epithelial cells (16HBE, HBE135-E6E7) were obtained from Fuhen Biologicals. The cells were cultured in KM medium (Sciencell, Cat#2101) supplemented with 1% keratinocyte growth supplement (KGS, Cat. No. 2152) and 1% penicillin-streptomycin (P/S, Cat. No. 0503). They were maintained in a humidified cell culture incubator at 37 °C with 5% $CO_2$. To induce cell injury, cells were treated with LPS (Sigma, Cat: L2630) at concentrations ranging from 1 to 800 μg/mL for 24 h. Moreover, cells were treated with various concentrations of the *CXCL1/CXCL2* inhibitor-Reparixin (0-40 μM, Selleck, Cat#S8640) for 24 h.

The CCK-8 assay is a widely used method for assessing cell proliferation and cytotoxicity. To perform the assay, 1×10^4 cells were plated in each well of a 96-well plate and then incubated for 24 h. Subsequently, LPS or Reparixin was added to six replicate wells in each group and the process was repeated 3 times. After 24 h of incubation, each well was treated with 10 μL of CCK-8 solution for 2 h (Abbkine, Cat#: BMU106-CN) and the optical density was measured at 450 nm.

After aspirating the culture solution from each well, we added 1 mL of Trizol to the cells in each well of the six well plate. The specific procedures and analytical methods were described above. The primer sequences were shown in Supplementary Table 3.

**Statistics and reproducibility.** The statistical analyses were calculated by R (version: 4.1.0, http://www.r-project.org) and GraphPad (Version: 7.0) software. The Shapiro-Wilk test was performed to confirm whether the data were normally distributed, and the Bartlett test was used for homogeneity of variances. Differences in immune infiltration between groups were analyzed using the Kruskal-Wallis test. Student's t-test was used for comparison between the two groups. Furthermore, Pearson correlation analysis was used to evaluate the relationship between the modules and clinical variables. The correlation between predicted genes and immune cells was assessed by Spearman correlation analysis. The *P*-value was corrected by the Bonferroni method and an adjusted *P*-value < 0.05 was considered statistically significant.

**Reporting summary.** Further information on research design is available in the Nature Portfolio Reporting Summary linked to this article.

## Results

**Association between asthma severity and neutrophil infiltration: insights from immune infiltration analysis.** To investigate the association between neutrophil infiltration and asthma progression, we conducted a preliminary quantitative analysis comparing immune cells infiltration in healthy individuals and asthma patients of varying severity. By analyzing the differential infiltration of 28 immune cells, ssGSEA results revealed that neutrophils (Kruskal-Wallis test, $P < 0.01$), mast cells (Kruskal-Wallis test, $P < 0.01$), plasmacytoid dendritic cells (Kruskal-Wallis test, $P < 0.01$), and regulatory T cells (Kruskal-Wallis test, $P < 0.05$) were higher in samples from SA patients than in healthy subjects and mild-moderate asthma patients (Fig. 2a).

Moreover, by using CIBERSORTx, ConsensusTME, MCP counter and xCell methods, we further confirmed statistically notable differences in neutrophil infiltration among the healthy controls, mild, moderate and SA groups (Fig. 2b–f, $P < 0.05$). Lung function indices (e.g., FEV1 and FVC) showed negative correlations with the degree of neutrophil infiltration (Fig. 2g, h, $P < 0.05$), providing additional evidence for the crucial role of neutrophils in asthma progression.

**Differential and enrichment analyses revealed molecular signatures affecting asthma progression.** To elucidate the molecular mechanisms underlying asthma progression, we conducted DEGs and functional enrichment analyses on gene expression profiles in asthma of different severities. Based on the above screening criteria, we analyzed DEGs in microarray data from healthy controls, mild, moderate and SA patients. Comparing mild asthma patients to healthy controls, we identified a total of 76 up-regulated and 37 down-regulated DEGs (Fig. 3a). Enrichment analysis revealed that these DEGs primarily participated in biological processes (BPs) such as "O-glycan processing" and "negative regulation of endopeptidase activity" (Fig. 3b). Remarkably, the "IL-17 signaling pathway" emerged as the only meaningful signaling pathway, with *MUC5AC*, *MUC5B* and *CXCL8* exhibiting impressive enrichment.

Between moderate and mild asthma, 42 up-regulated and 103 down-regulated DEGs were identified (Fig. 3c). Enrichment analysis revealed that "negative regulation of substrate adhesion-dependent cell spreading" was the most remarkable BP (Fig. 3d). KEGG analysis showed that the "HIF-1 signaling pathway" was the most important signaling pathway, with TFRC, NOS2 and ERBB2 enriched in this pathway. ERBB2 (HER2), the second member of the EGFR family[29], is also involved in epithelial repair in asthmatic patients.

Between severe and moderate asthma, 293 up-regulated and 289 down-regulated DEGs were identified (Fig. 3e). BPs analysis showed that "antigen processing and presentation of endogenous

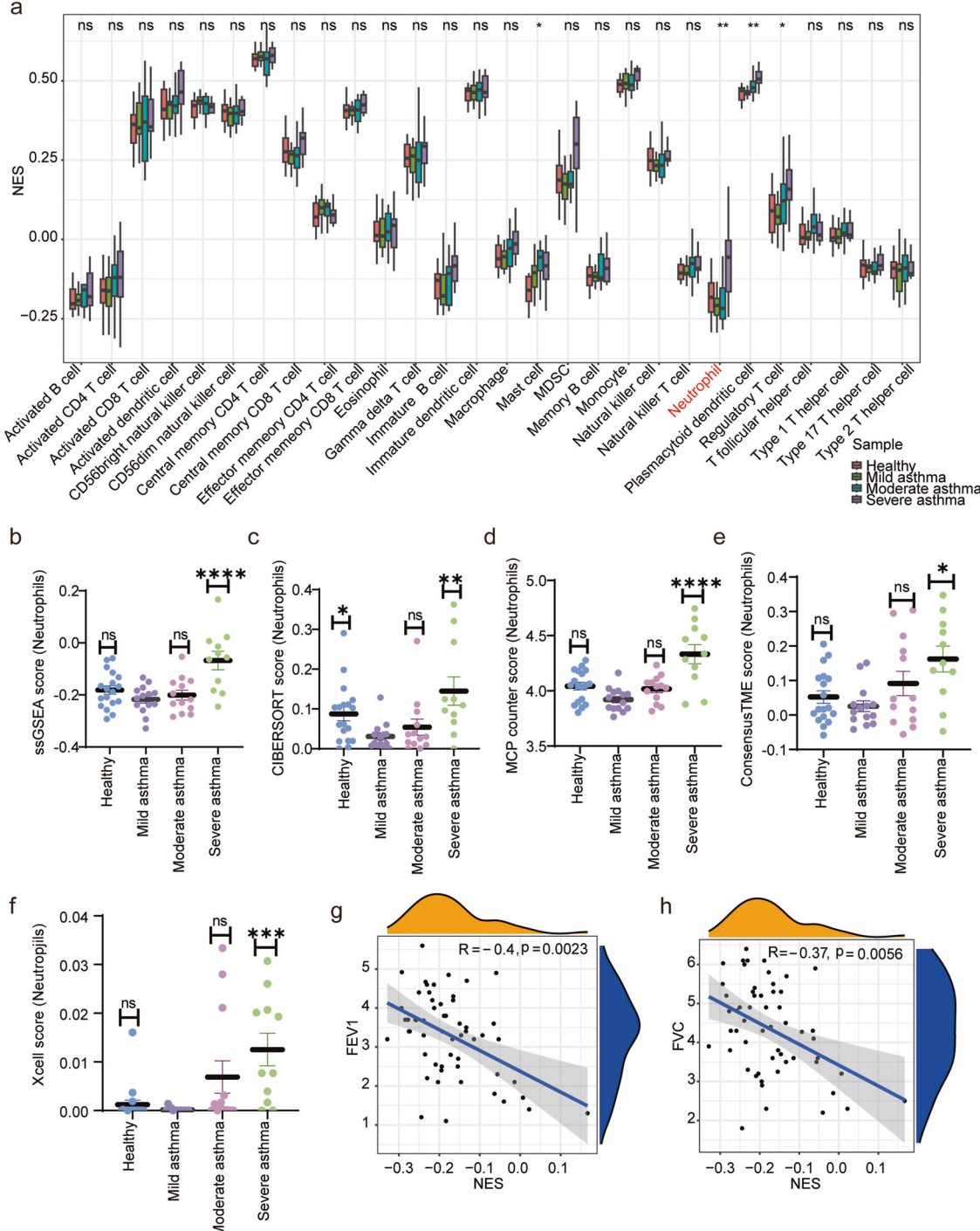

**Fig. 2 Immune infiltration analysis. a** Scatterplot of the extent of 28 immune cell infiltration; NES: Normalized Enrichment Score; **b–f** The degree of neutrophil infiltration in the healthy, mild, moderate and severe groups was calculated using ssGASE, CIBERSORT, ConsensusTME, MCP counter and X-cell methods, respectively. ns: $P > 0.05$; *$P < 0.05$; **$P < 0.01$; ***$P < 0.001$; ****$P < 0.0001$. **g, h** Spearman correlation analysis of FEV1, FVC and the degree of neutrophil infiltration was performed using the ssGSEA method. Error bars indicate mean ± SEM.

antigen" were the most important BPs (Fig. 3f). KEGG analysis showed that "Cell adhesion molecules", "Longevity regulating pathway", and "Cellular senescence" were significantly enriched.

Based on the aforementioned results, we explored in greater depth the molecular features that influence asthma progression. We identified 259 up-regulated and 604 down-regulated DEGs between SA and mild asthma patients (Fig. 3g). BPs analysis indicated the highest enrichment of "neutrophil degranulation"

and "neutrophil activation involved in immune response", which also included processes such as "neutrophil chemotaxis" and "neutrophil migration" (Fig. 3h). Furthermore, KEGG enrichment analysis highlighted the significant enrichment of pathways such as "Osteoclast differentiation" and "Viral protein interaction with cytokine and cytokine receptor". Consequently, a total of 863 DEGs were detected between SA and mild asthma for further analysis.

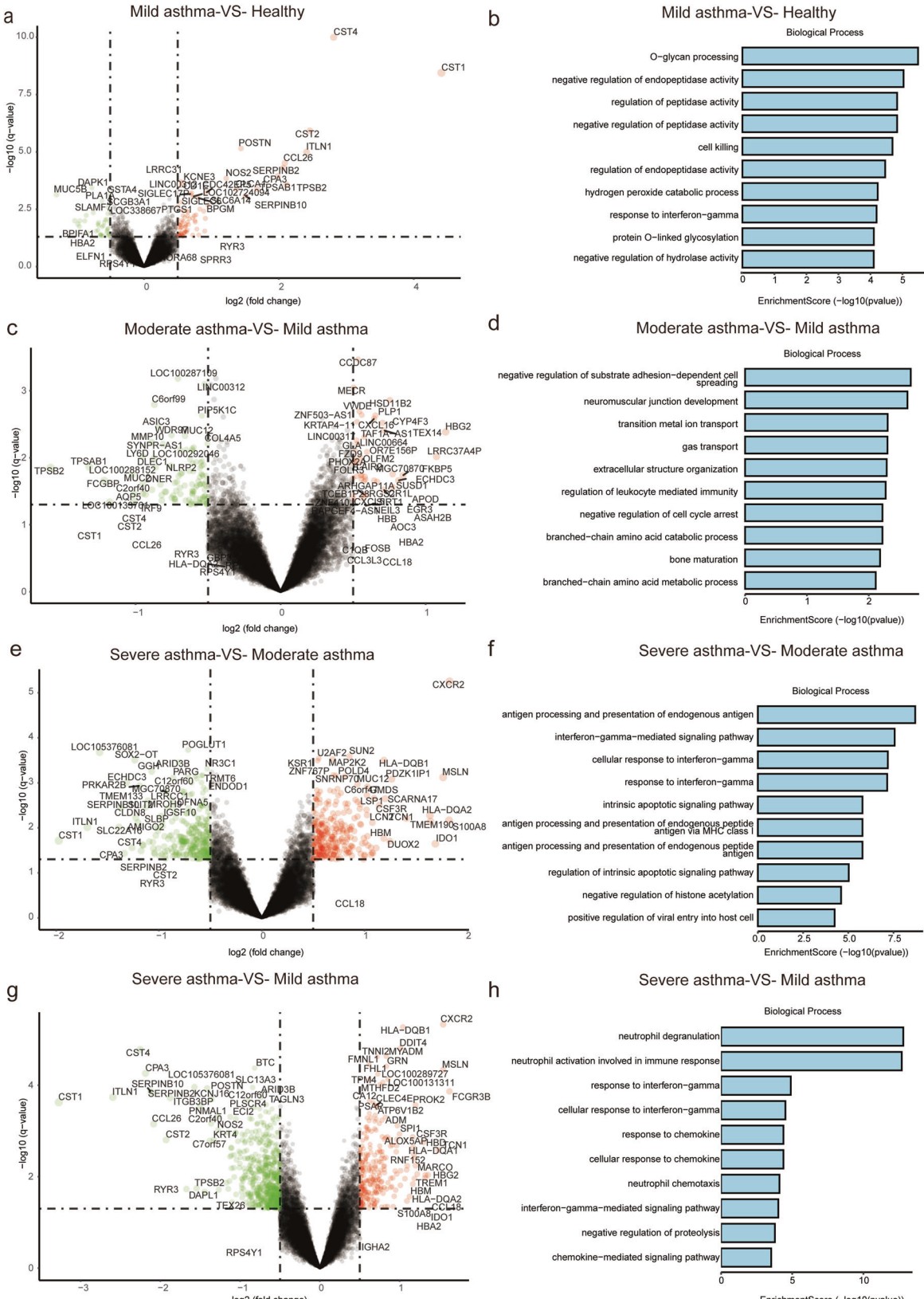

**Fig. 3 Differential gene analysis between healthy controls, mild, moderate, and severe asthma. a**, **b** Volcano plot between mild asthma group and healthy controls, and the enrichment analysis of biological processes. **c–h** Volcano plots between moderate and mild asthma patients, severe and moderate asthma patients, and severe and mild asthma patients, respectively, alongside the corresponding bioprocess enrichment analysis.

**Modules associated with asthma severity and neutrophil infiltration identified by co-expression network analysis**. To confirm co-expression modules related to neutrophils and asthma severity, 1793 immune genes were merged with the GSE89809 matrix to obtain expression data for immune-related genes. Based on the expression of 1318 genes in GSE89809, the signed networks were analyzed using the R package "WGCNA" to identify the co-expression modules for asthma. First, cluster analysis identified 6 free samples, and $\beta = 7.4$ co-expression modules were identified in the scale-free co-expression network with R^2 > 0.85. The turquoise module showed the strongest relationship with SA (correlation coefficient = −0.53, P = 0.002) and neutrophils (correlation coefficient = 0.75, P = 7e − 07). In addition, there was a positive correlation between the turquoise module and asthma control questionnaire (ACQ) (correlation coefficient= 0.46, P = 0.008), inhaled corticosteroid (ICS) dose (correlation coefficient = 0.52, P = 0.002) and Global Initiative for Asthma (GINA) control (correlation coefficient = 0.51, P = 0.003). Apart from the gray modules, other modules had some specific asthma symptoms. Therefore, the turquoise module was considered to play an important role in the progression of asthma. Finally, we obtained a total of 328 genes in the turquoise module (Fig. 4).

**Key targets involved in asthma severity and neutrophil infiltration identified by GGI and PPI networks**. To investigate the pivotal targets associated with neutrophil activation during asthma development, we cross-analyzed 863 DEGs observed in SA and mild asthma with 328 targets identified by WGCNA. This analysis identified 46 common targets, and they were entered into the STRING database to construct the PPI network. It turned out that 46 nodes and 240 edges had minimum interaction scores greater than 0.4 (Fig. 5a). Furthermore, it was necessary to filter the critical nodes of the turquoise module by "CytoHubba" plugin in Cytoscape software, and the 10 critical genes were PTPRC, TLR2, MMP9, FCGR3B, TYROBP, CXCR1, S100A12, FPR1, CCR1 and CXCR2 (Fig. 5b).

GeneMANIA was used to construct a GGI network to predict the potential interactions between these 10 genes and other targets. The results showed that the top 10 hub genes interacted with S100A8, S100A9, LILRB3, FCER1G, MCER1G and MMP25. These genes were mainly related to myeloid leukocyte migration (FDR = 2.27e-11), immune receptor activity (FDR = 2.27e-11), leukocyte chemotaxis (FDR = 6.91e-10), neuroinflammatory response (FDR = 1.72e-5), neutrophil migration (FDR = 4.92e-9), regulation of inflammatory response (FDR = 4.39e-5) and cell chemotaxis (FDR = 5.85e-11) (Fig. 5C).

**Enrichment analysis yielded biological processes associated with asthma progression**. To further investigate the underlying molecular mechanisms of asthma progression, we selected 46 genes from the turquoise module and DEGs between SA and mild asthma for subsequent enrichment analysis. Enrichment analyses showed that genes were significantly enriched in leukocyte and neutrophils chemotaxis, cell chemotaxis, cytokine-mediated signaling pathway, myeloid leukocyte migration, neutrophil migration and granulocyte chemotaxis (Fig. 6a and Supplementary Table 4). In addition, they were intimately involved in cytokine-cytokine receptor interaction, viral protein interaction with cytokine and cytokine receptor, chemokine signaling pathway, phagosome, rheumatoid arthritis, rndocytosis, neutrophil extracellular trap formation, and PI3K-Akt signaling pathway, all of which contributed to regulating the host immunity (Fig. 6b and Supplementary Table 5).

**Analysis of upstream pathways activities and TF-critical gene regulatory network associated with asthma progression**. To infer upstream pathway activity from the expression of 46 DEGs, the genes were input into the SPEED2 online website. SPEED2 quantified the genes in the gene list to determine whether characteristic genes were enriched in strongly dysregulated pathways. As shown in Fig. 6, the darker the color, the more significant the false discovery rate (FDR), with PPAR, IL-1 and MAPK + PI3K featured prominently in our findings (Fig. 7a).

To explore the effect of the key genes in-depth, two TF-gene regulatory networks were assembled. According to the screening criteria of sub-networks with the number of nodes greater than 3, the final regulatory networks and detailed information of key genes and TFs were shown in Fig. 7b and Supplementary Table 6. One of the sub-networks included 5 hub genes, and there were 109 nodes, 114 edges and 5 seeds in the TF-gene network. Another one had 2 seeds, 5 nodes and 4 edges, suggesting the potential regulatory roles for core genes and TFs. It is worth noting that ARID1B was the most connected TF with 3 targets including TYROBP, FPR1 and FCGR3B

**Correlation analysis between key genes and immune cells**. We conducted a Spearman correlation analysis of the relationships between 10 key genes and 28 immune cells. The results revealed a statistically significant correlation between CXCR2 (correlation coefficient = 0.73, P-value = 2.53E-10) and neutrophils. All genes except TLR2 and TYROBP were associated with neutrophils ($p < 0.05$). Notably, S100A12 showed the strongest correlation with neutrophils (correlation coefficient = 0.57, P-value = 5.31E-06), whereas MMP9 displayed moderate correlation (correlation coefficient = 0.32, P-value = 1.51E-02).

Furthermore, CXCR1 exhibited the strongest correlation with activated regulatory T cells (correlation coefficient = 0.76, P-value = 1.19E-06), as did CCR1 (correlation coefficient = 0.79, P-value = 5.44E-13). FCGR3B demonstrated the highest correlation with central memory CD8 + T cells (correlation coefficient = 0.78, P-value = 1.74E-12), while FPR1 exhibited the strongest correlation with MDSC (correlation coefficient = 0.70, P-value = 2.18E-09). TYROBP had the strongest correlation with activated dendritic cells (correlation coefficient = 0.80, P-value = 9.15E-14), and TLR2 displayed the highest correlation with immature dendritic cells (correlation coefficient = 0.52, P-value = 3.82E-05). Lastly, PTRPC showed the strongest correlation with MDSC (correlation coefficient = 0.78, P-value = 1.89E-12). It is noteworthy that each target exhibited associations with multiple immune cells (Fig. 8).

**Identification of key drugs to block the progression of asthma**. The CTD database was used to predict potential drugs that may intervene or treat asthma. According to enrichment FDR, 2-(4-isobutyl-phenyl)-propionyl-methane-sulfonamide (Reparixin) ranked highest in our analysis, which targeted MMP9, CXCR1 and CXCR2. Of note, Reparixin is an inhibitor of CXCR1 and CXCR2 with potential antineoplastic activity. Cytochalasin D and Methotrexate may play a vital role in asthma (Supplementary Table 7).

**Validating key targets of NA in an in vivo model**. On the basis of previous studies, we developed a mouse model using OVA and CFA to validate the key pathological manifestations and expression of key targets in NA. The corresponding flowchart was shown in Fig. 9a. HE staining showed a large infiltration of inflammatory cells around the trachea, bronchus and pulmonary artery in NA mice compared with controls (Fig. 9b). PAS staining

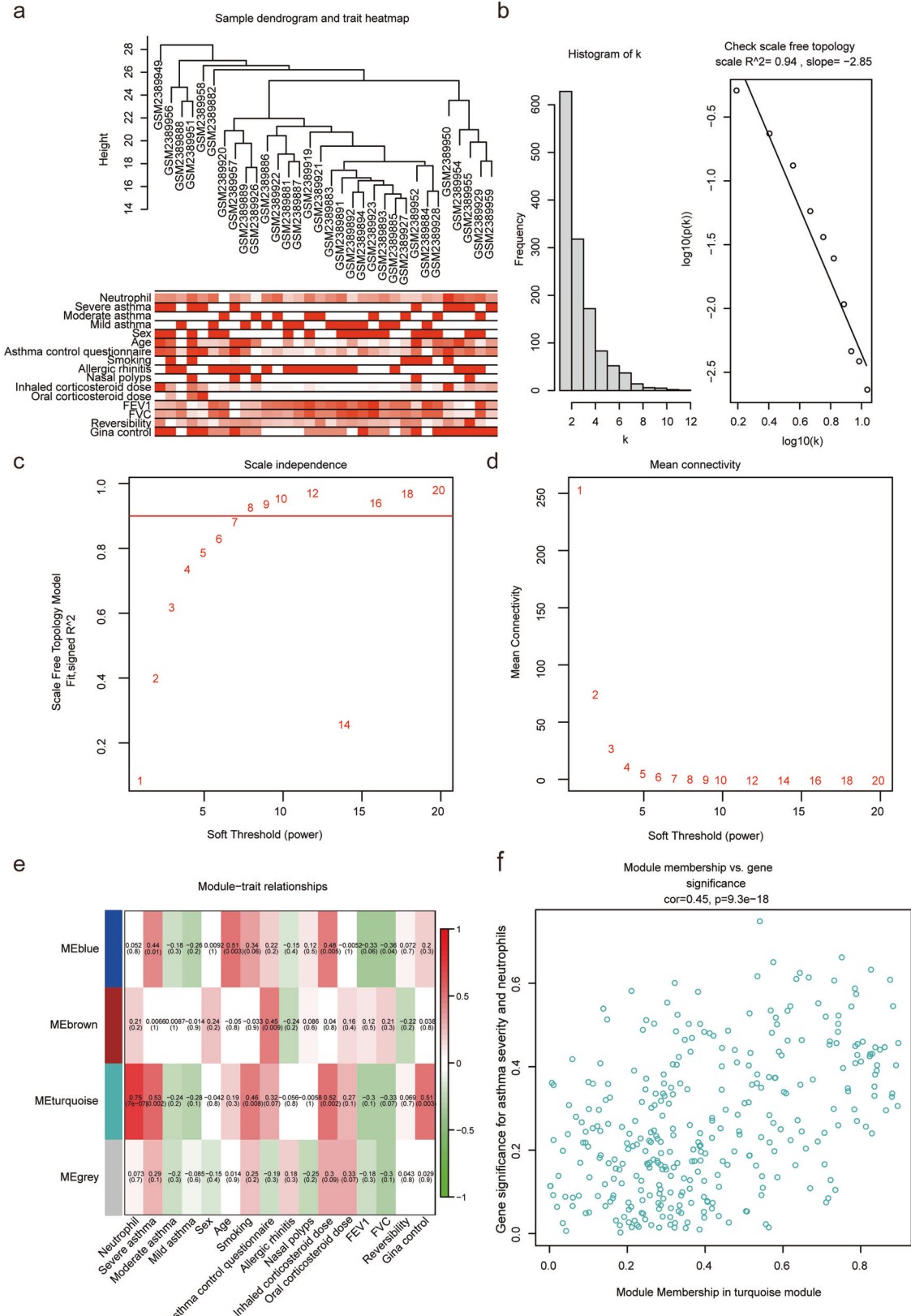

confirmed the presence of goblet cell hyperplasia and excessive mucus production in NA mice compared to control group (Fig. 9c). Notably, collagen deposition was significantly increased in the lungs of NA mice compared to controls (Fig. 9d).

Wright-Giemsa staining of BAL revealed a substantial increase in the total number of leukocytes, especially neutrophils, in NA

mice (Fig. 9e). We further analyzed neutrophil-related cytokines in BAL and serum using ELISA. The expression of MPO, a marker of neutrophils, was significantly increased in NA mice compared to controls. Moreover, the levels of IgE, IL-1$\beta$, TNF-$\alpha$, MPO and IL-8 were elevated in serum or BAL in the NA group (Fig. 9f). Additionally, we observed that mRNA levels of several

**Fig. 4 Construction of weighted gene co-expression network. a** Sample dendrogram and trait heatmap sample cluster analyses; **b** Determination of soft-threshold power in WGCNA. The histogram of connectivity distribution and scale-free topology were shown in the first two panels, respectively; **c, d** Scatter plots of the mean connectivity and the value of the soft threshold (power). The abscissa was the power value and the ordinate was the scale-free fit index. The higher the value of the ordinate, the more the network fit the scale-free feature. **e** Identification of the modules related to asthma severity and neutrophils. The horizontal axis represented different clinical factors and the vertical axis represented different modules. The correlation coefficient and *P* value were displayed in the module. Red color indicated positive correlation, while green color indicated negative correlation. The darker the color, the stronger the correlation. FEV₁ forced expiratory volume in the first second, FVC forced vital capacity, GINA Global Initiative for Asthma. **f** Correlation between module membership in turquoise module and gene significance for asthma severity and neutrophils.

inflammatory genes associated with neutrophils and disease progression (i.e., *PTPRC, CCR1, FPR1, TLR2, MMP9, CXCR1, CXCR2, TYROBP* and *FCGR3B*) were up-regulated in NA mice (Fig. 9g). Since S100A12 is not present in mice, we did not validate its mRNA levels in vivo[30].

**Reparixin inhibited CXCR1, CXCR2 and MMP9 expression in LPS-stimulated human 16HBE cells.** Figure 10a illustrated the 2D structures of Reparixin. As shown in Fig. 10b, exposure to LPS resulted in a dose-dependent decrease in 16HBE cell survival, which decreased by 50% at a concentration of 480.8 µg/mL (N = 3, IC50 = 480.8 µg/mL). Therefore, LPS at a concentration of 480.8 µg/mL was used for subsequent studies. To assess the cytotoxicity of Reparixin on 16HBE cells, cells were treated with concentrations of Reparixin ranging from 0 µM to 40 µM for 24 h. The results revealed that Reparixin did not inhibit cell viability at concentrations ranging from 0 to 40 µM (Fig. 10c). Furthermore, in order to further understand the role of Reparixin in bronchial epithelial cells, we evaluated its effect on LPS-injured 16HBE cells. LPS-injured 16HBE cells incubated with 1 µM, 2 µM and 4 µM concentrations of Reparixin showed a dose-dependent increase in cell survival (Fig. 10d), indicating that Reparixin exhibited a protective effect on 16HBE cells. Finally, RT-qPCR analysis confirmed that the expression of 9 key targets in addition to PTPRC was also elevated in LPS-damaged 16HBE cells. Importantly, Reparixin was able to reduce the expression levels of *CXCR1, CXCR2* and *MMP9* (Fig.10e, f).

## Discussion

NA is a clinical term characterized by frequent deterioration[31] and often results in varying degrees of lung function impairment[32]. Despite appropriate therapies, a proportion of patients have persistent symptoms of airflow obstruction[33,34]. As a result, more targeted therapeutic interventions are needed. Innate and adaptive immune responses play a critical role in the inflammatory regulation of asthma[7]. Therefore, identifying biomarkers, molecular mechanisms and differences in immune infiltration related to asthma severity is crucial for predicting targets and possible therapeutic drugs.

However, neutrophils will release chemokines that attract monocytes into the airway to alter airway inflammatory profile[35]. Meanwhile, neutrophils also can cause airway hyperresponsiveness in humans[36]. On this basis, our study used bioinformatics approaches to confirm the key modules closely related to neutrophils and asthma severity. Also, immune infiltration mechanisms and targets that may cause asthma deterioration were investigated the by GGI and PPI networks. We finally obtained 10 critical genes that may exacerbate asthma by promoting neutrophil infiltration and predicted potential treatments. We performed in vitro and in vivo experiments to verify this hypothesis. First, we successfully established an NA mouse model by OVA and CFA. Wright-Giemsa staining of BAL and ELISA assay of BAL and serum indicated that neutrophil-related indicators were significantly increased in NA mice. HE, PAS and

Masson staining showed a large infiltration of inflammatory cells, excessive mucus production and increased collagen deposition in NA mice compared to controls. All the above results verified that NA was successfully induced by OVA and CFA in vivo. Then, 16HBE cells were exposed to LPS to induce a model of bronchial epithelial cell injury in vitro. We finally verified the transcript levels of the 10 core genes and simultaneously predicted potential drugs using in vivo and in vitro models.

Our results implied that immature mast cells, neutrophils, plasmacytoid dendritic cells, plasmacytoid dendritic cells and regulatory T cells were more abundant in SA samples than in healthy individuals and mild-moderate asthma samples.

Mast cells occupy a unique position among immune response cells and contribute to innate and adaptive immunity[37]. Mast cell activation will synthesize and release cytokines, chemokines, and growth factors, which trigger a programmed inflammatory response[38–40]. Mast cell-derived mediators induce the classical features of the early asthmatic reaction, inducing bronchoconstriction, mucus secretion and mucosal oedema[41,42]. Thus, mast cells play an important role in the asthma response.

Dendritic cells are antigen-presenting cells located at the core of the innate immune system and are highly sensitive to the environment[43]. The activation state of dendritic cells is essential for the differentiation of Th1 and Th17 cells. Heleen Vroman et al. show that inhibition of dendritic cells activation may have a certain therapeutic effect on TH17-mediated neutrophil inflammation in SA patients, which may be a novel pharmacological intervention[44].

Regulatory T cells (Tregs) are cells that can suppress or regulate lymphocytes and, as such, prevent immune responses to self-antigens and maintain immune homeostasis[45]. CD8+ Tregs are the first reported subset of Tregs with immunosuppressive functions[46,47]. A single-cell sequencing study reveals that CD8 + T cell significantly increases in the BAL of SA patients[48]. Eleni Tsitsiou et al. also indicate that SA is involved in activating the circulating CD8 + T cells[49]. It was confirmed that activated CD8 + T cells are on the rise in postmortem lung tissue samples from patients who die of acute asthma[50].

The most common manifestation of SA is airway neutrophilic inflammation, with neutrophils contribute to the activation of inflammatory pathways leading to NA deterioration[51,52]. In addition, the presence of autophagy and extracellular trap in peripheral neutrophils exacerbate asthma severity, triggering a cascade of inflammatory responses in airway epithelial cells[53]. Clinical cluster analysis of the SA Research Program defines 5 asthma subtypes by clinical cluster analysis, which shows a high correlation between sputum neutrophil count and SA phenotype[54].

Therefore, we analyzed the degree of neutrophil infiltration among healthy controls, asthma patients of different severity through various methods. Consistent with previous findings[55], asthma severity was closely related to the infiltration of neutrophils. Furthermore, neutrophils were inversely proportional to FEV1 and FVC, further indicating the value of neutrophils in assessing asthma severity. In conclusion, our results based on the

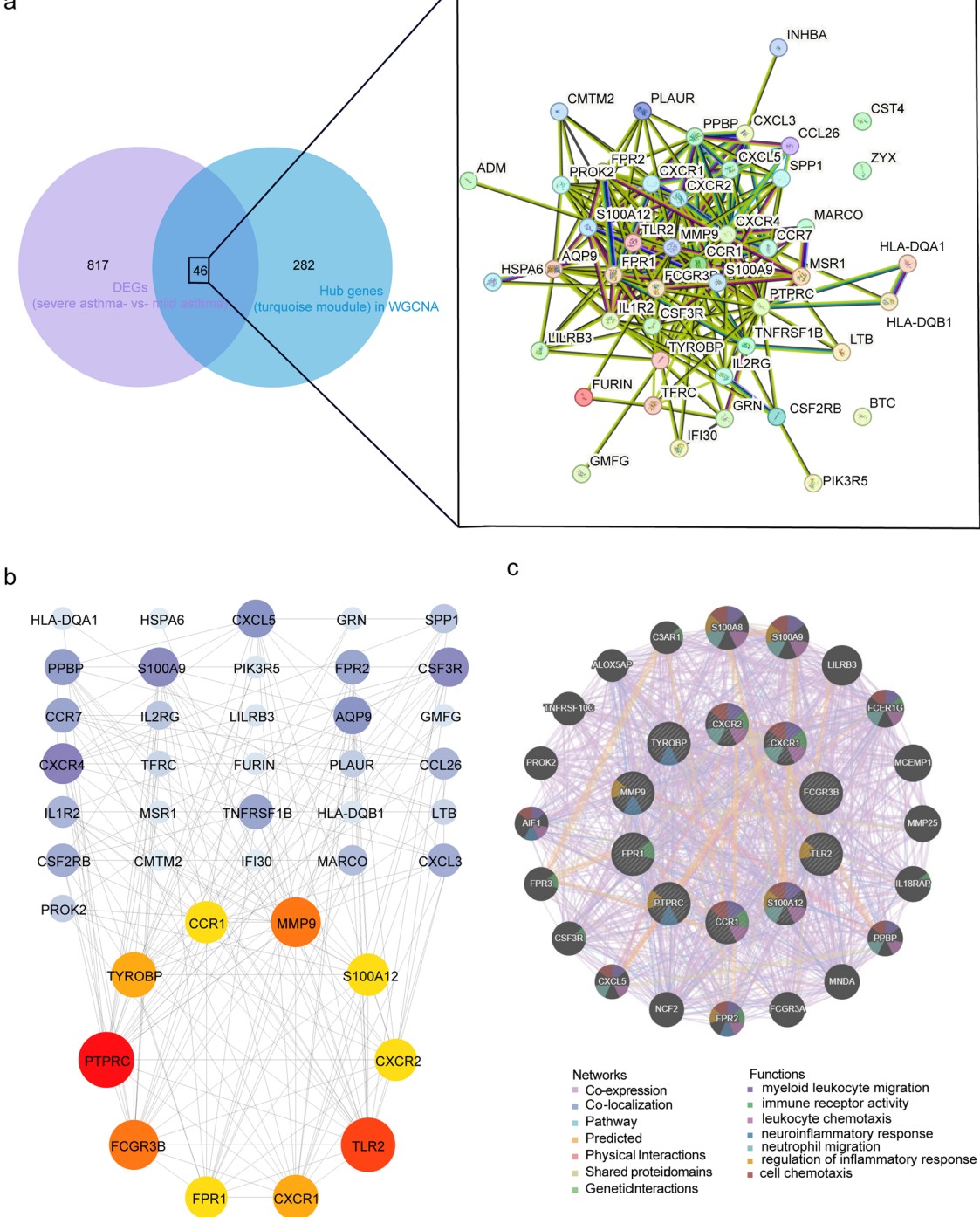

**Fig. 5 Construction of protein–protein interaction (PPI) and gene–gene interaction (GGI) networks of the top 46 genes. a** Venn diagram on the left and PPI network on the right in the STRING database. The lines represented the protein–protein interactions. **b** Network diagram of 46 gene interactions. The nodes below were the top 10 key genes with the highest degree in different colors. The darker the color, the higher the degree of genes. **c** The GGI networks and functions between genes were shown in different colors. The genes in the center circle were the 10 key targets, and the genes in the outer circle were the genes with the strongest interactions with these 10 genes. Different colors represented different network modes and functions.

assessment of immune cell infiltration of airway epithelial cell transcriptome are in line with the current research. However, the need to further understand the immune cell infiltration mechanism in asthma severity is still in urgent need.

Using the WGCNA from 6 samples in GSE89809, we constructed a scale-free network with $\beta = 7.4$ and found that the turquoise module was most relevant to the asthma severity and

neutrophils. Moreover, the turquoise module was positively correlated with ACQ, ICS dose and GINA control. Therefore, further analysis of the biological functions and mechanisms of the genes in the turquoise modules has vital research value.

Thus, we selected 46 genes from the turquoise module and DEGs between SA and mild asthma to investigate the underlying molecular mechanisms. Enrichment analyses showed that the

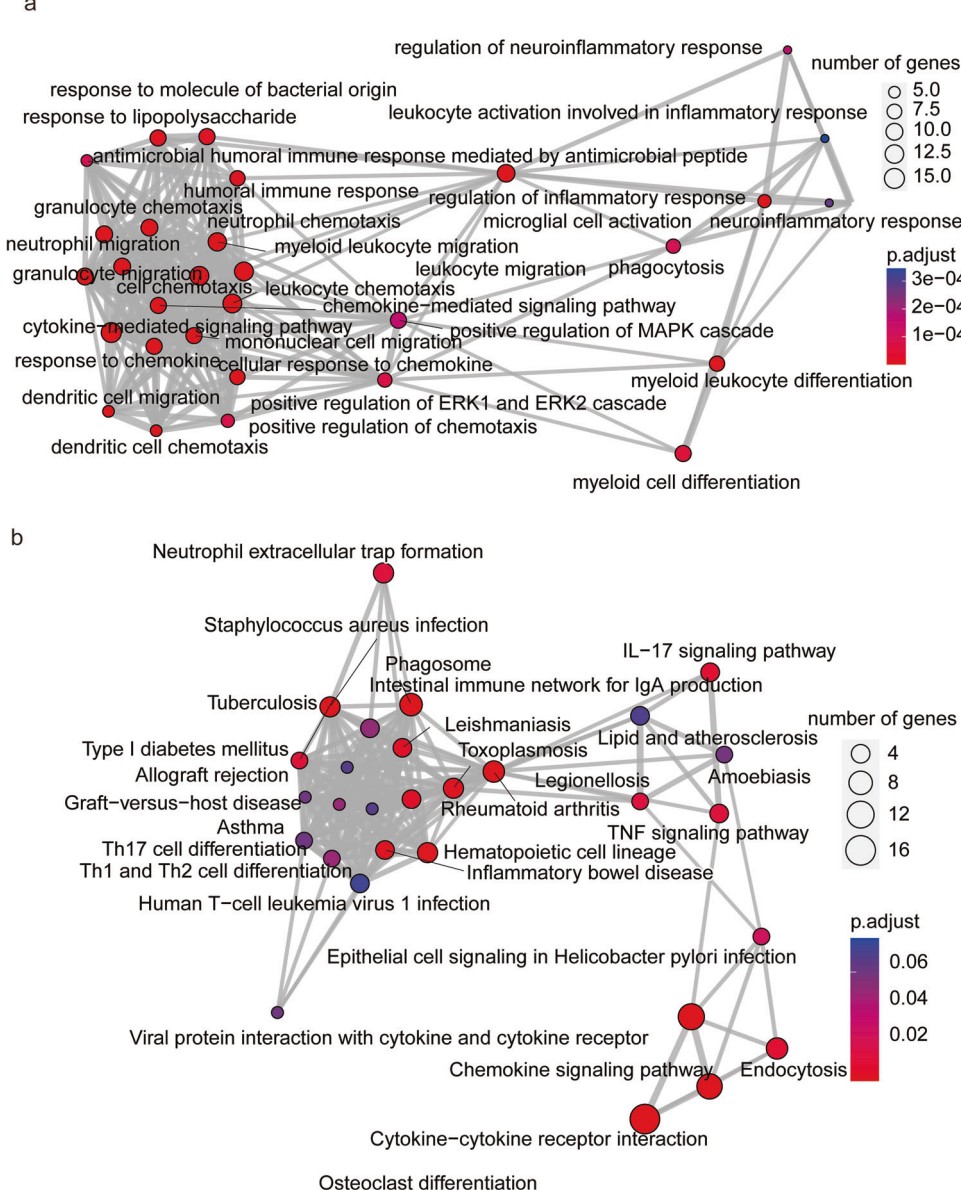

**Fig. 6 Enrichment analysis of top 46 genes.** Top 30 terms of biological process (**a**) and top 20 Kyoto Encyclopedia of Genes and Genomes (**b**) pathway enrichment analyses. The size of the dot represented the number of enriched genes, the color of the dot represented the P value and the line represented the interaction between the items.

turquoise module was mainly enriched in leukocyte and neu-trophils chemotaxis, cell chemotaxis, cytokine-mediated signaling pathway, myeloid leukocyte migration, neutrophil migration and granulocyte chemotaxis. In addition, they were intimately involved in cytokine-cytokine receptor interaction, viral protein interaction with cytokine and cytokine receptor, chemokine sig-naling pathway, phagosome, rheumatoid arthritis, endocytosis, neutrophil extracellular trap formation, and PI3K-Akt signaling pathway, all of which contributed to regulating the host immu-nity. Cytokine-cytokine receptor interactions, viral protein interactions with cytokines and cytokine receptors and chemo-kine signaling pathways are important in asthma pathogenesis.

By comparing the enriched signaling pathways in mild asthma and healthy controls, we identified "IL-17 signaling pathway" as the only important signaling pathway in which *MUC5AC*, *MUC5B* and *CXCL8* showed significant enrichment. IL-17, also known as IL-17A, has a central role in asthma. IL-17 is highly expressed in induced sputum and bronchial biopsy of SA patients, and is associated with neutrophil airway inflammation and airway remodeling (AR)[56]. IL-17/IL-23 cytokines can upregulate the expression of glucocorticoid receptors β and may play a role in steroid resistance in SA patients[57]. The activation of inflamma-tory pathways in different asthma endotypes mostly involves various cytokines and growth factors. In a prospective study[58], plasma levels of CXCL8 are increased in asthma individuals compared to healthy controls. Our findings showed that HE staining revealed a large infiltration of inflammatory cells around the trachea, bronchus and pulmonary artery in NA mice com-pared to controls. This result described above demonstrated the existence of airway inflammation. According to research on fatal asthma, the major cause of death in asthma is airway lumenal obstruction by mucus[59,60]. MUC5AC and MUC5B are the major gel-forming mucins that are abundantly produced in the intra-pulmonary airways[61].

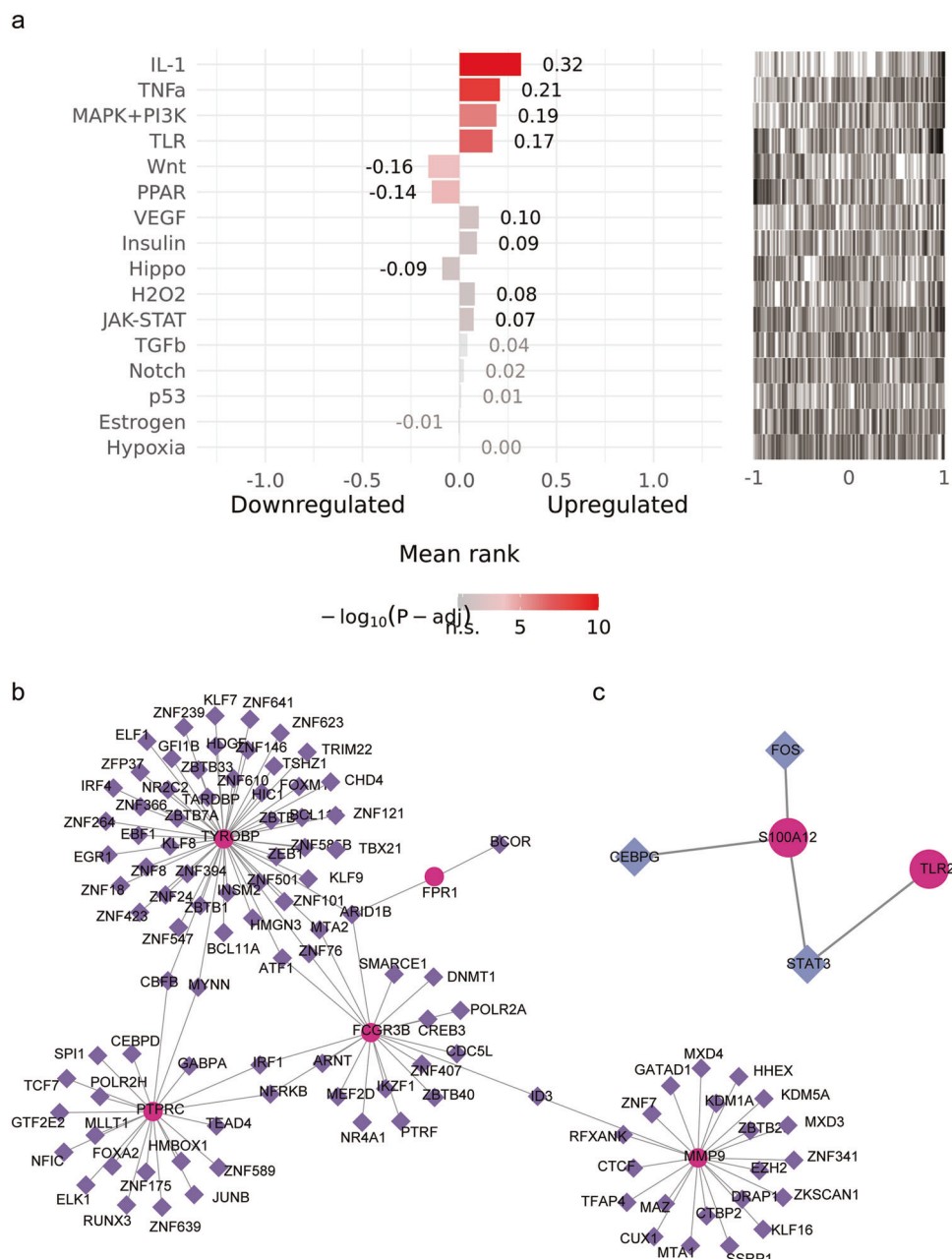

**Fig. 7 The activity level of the upstream signaling pathway of the 46 targets in the turquoise module (a, adjusted *P*-value < 0.05) and the interaction network of 10 identified genes with transcription factors (TFs) (b-c).** Red circles represented the candidate gene, purple triangles represented the transcription factor and lines represented the interaction between the candidate gene and the TF.

Between moderate and mild asthma, KEGG analysis showed that the "HIF-1 signaling pathway" was the most significant signaling pathway, with TFRC, NOS2 and ERBB2enrichment. ERBB2 (HER2), the second member of the EGFR family[29], is also involved in epithelial repair in asthmatic patients. HIF-1 is a transcription factor downstream of EGFR[62], and EGFR expression is up-regulated in the airway epithelium of asthmatics[63,64]. Young HW et al. demonstrate that the HIF-1 cis motifs are required for induction of the MUC5AC promoter by IL-13 or EGF[62]. HIF-1 can acutely response to a various of inflammatory signals[65,66], and thus may be a central regulator for MUC5AC overproduction and mucous metaplasia. Notably, MUC5AC transcription level is correlated to the percentage of sputum neutrophils[67]. In our study, PAS staining confirmed the presence of goblet cell hyperplasia and excessive mucus production.

Therefore, the results of the in vivo experiments corroborate with the bioinformatics analysis described above. We explored in more depth the molecular features that influence the asthma progression. BPs analysis indicated the highest enrichment of "neutrophil degranulation" and "neutrophil activation involved in immune response", which also included the processes of "neutrophil chemotaxis" and "neutrophil migration". Therefore, neutrophil played an important role in the development of asthma.

Airway inflammation and AR are important features of asthma. AR includes structural changes in airway wall tissues and occurs concurrently with airway inflammation. The relationship between airway inflammation and AR is complicated and the two can interact with each other[7]. Furthermore, the role of neutrophils in AR has been increasingly highlighted in asthma[68]. Previous studies in mouse models have demonstrated that inhibition of HIF reduces

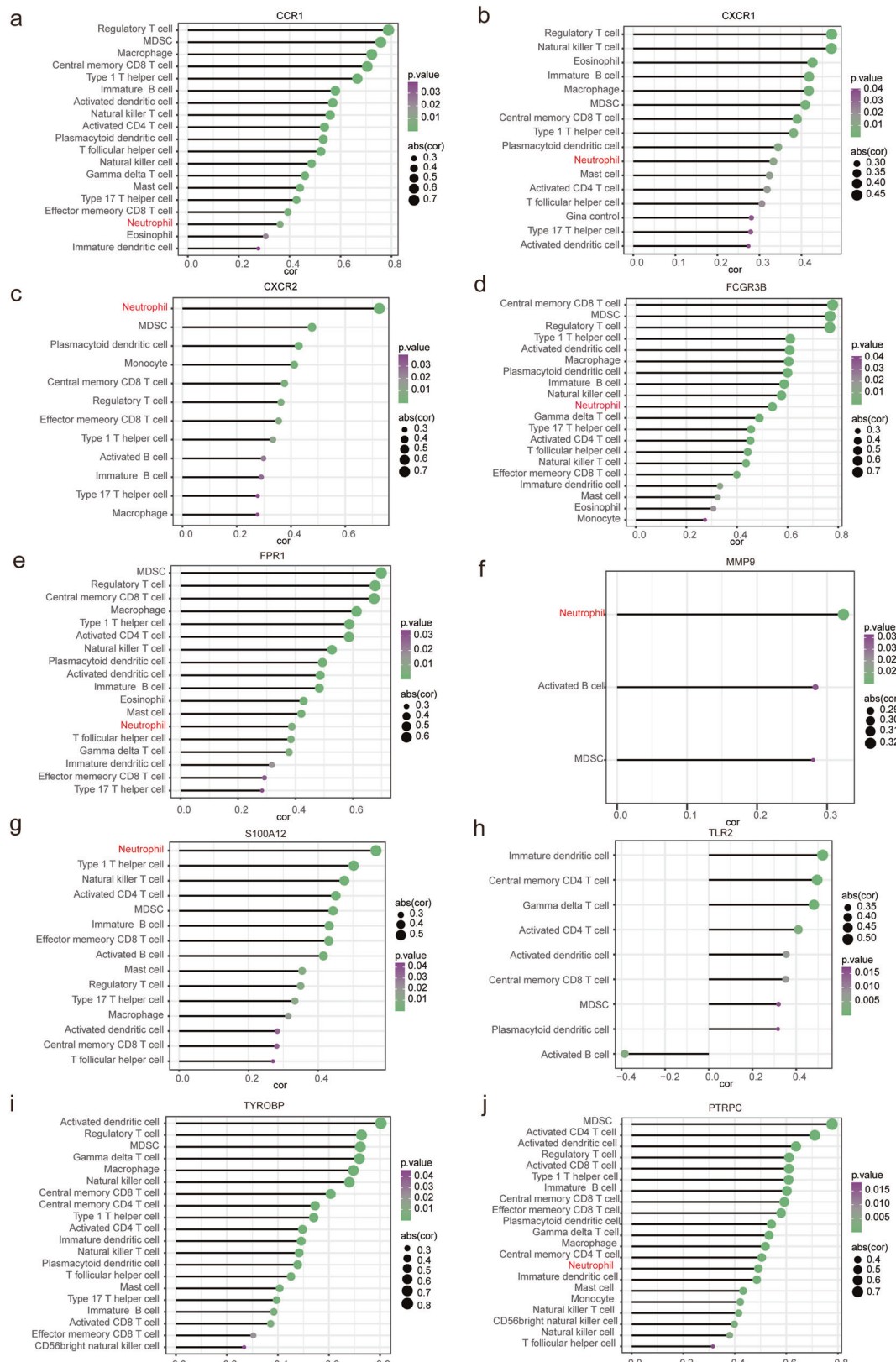

**Fig. 8 Spearman correlation analysis between 28 immune cell types and 10 hub genes.** The genes represented in (**a**)–(**j**) were CCR1, CXCR1, CXCR2, FCGR3B, FPR1, MMP9, S100A12, TLR2, TYROBP and PTRPC. The horizontal axis represented the absolute value of the correlation coefficient, and the vertical axis represented the immune cells. The color of the point reflected the size of the *P*-value, and the size of the point reflected the absolute value of the correlation coefficient.

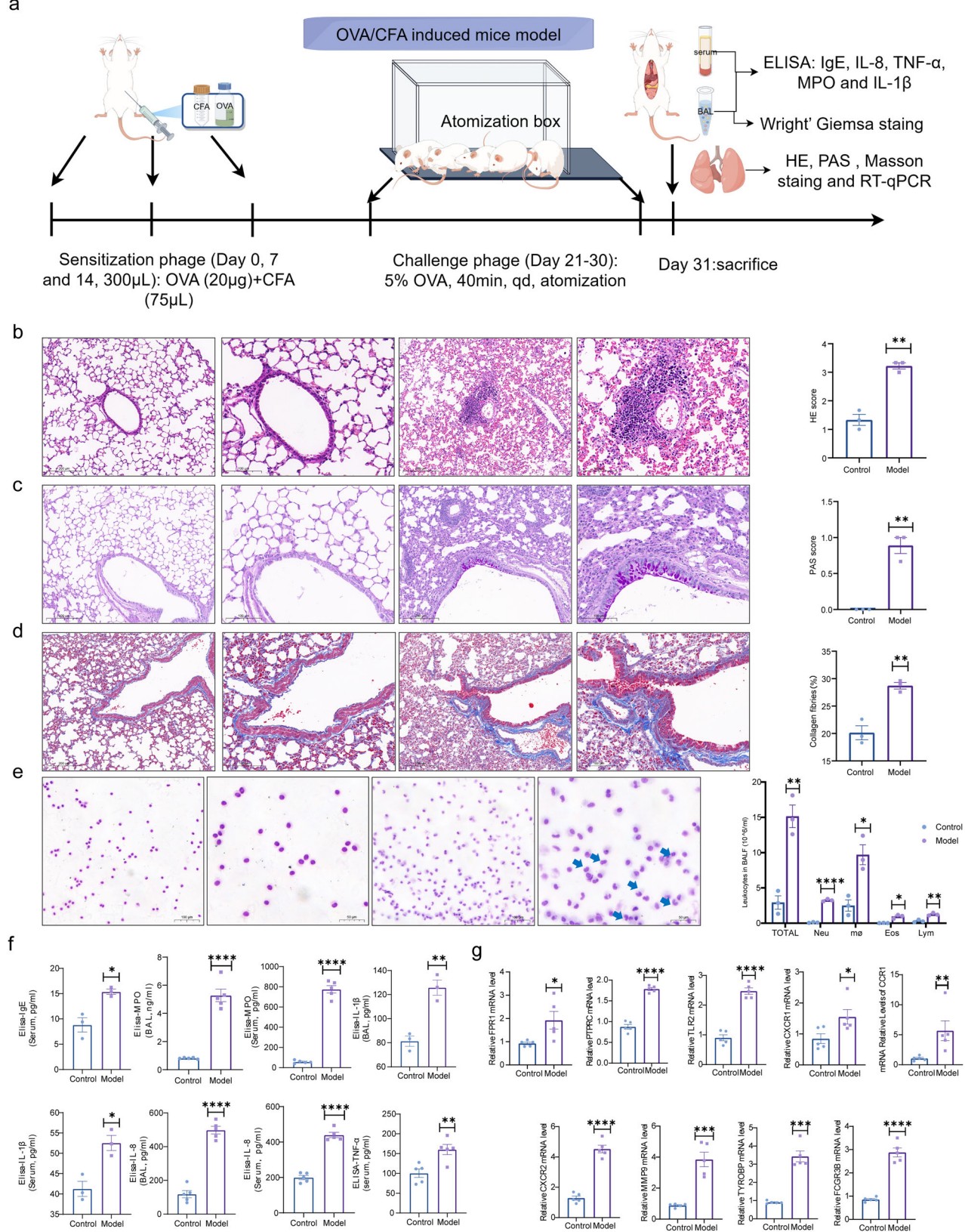

inflammation and remodeling under normoxic conditions[69,70]. On the other hand, chronic administration of azithromycin has a protective effect on AR, and this protective effect is accompanied by a remarkable downregulation of HIF-1α[71].

Taken together, neutrophil activation has the effect of initiating AR and accelerating lung function decline. IL-17 and HIF-1 signaling pathways show the dual therapeutic potential to modulate airway inflammation and AR on asthma severity. Notably, collagen deposition was significantly increased in the lungs of NA mice compared to controls, confirming the effect of neutrophils on AR.

It has been shown that in addition to regulating gene expression, TFs are also strongly associated with the onset and

**Fig. 9 Construction of an OVA/CFA-induced asthma mouse model. a** Flowchart of mouse model establishment; **b–d** HE, PAS and Masson's trichrome staining of lungs from control and model groups. Scale bars were 100 $\mu$M and 20 $\mu$M. The respective pathology scores were summarized in the right panel (N = 3). **e** Wright-Giemsa staining of bronchoalveolar lavage (BAL) fluid. Scale bars were 100 $\mu$M and 50 $\mu$M. The graph on the right showed the total cell counts and four different cell counts (neutrophils, eosinophils, lymphocytes and macrophages) in the BAL, respectively (N = 3), and the blue arrow represented neutrophils. **f** Total IgE level in serum (N = 3), MPO (N = 5), IL-1$\beta$ (N = 3) and IL-8 (N = 5) levels in serum and BAL, and TNF-$\alpha$ (N = 5) level in serum were measured by ELISA; **g** RT-qPCR analysis of hub genes by $2^{-\Delta\Delta Ct}$ method (N = 5). *$p < 0.05$; **$p < 0.01$; ***$p < 0.001$; ****$p < 0.0001$. Error bars indicate mean ± SEM.

progression of diseases[72]. In our research, we found that *ARID1B* was the TF with the highest association with 3 targets, including *TYROBP*, *FPR1* and *FCGR3B*. Mutations in *ARID1B* are associated with increased TMB in various cancers[73,74]. In contrast, the mutations of *ARID1B* may influence the percentage of M1 macrophages, T helper cells, resting memory CD4 + T cells, monocytes and activated dendritic cells[74].

TYROBP, a transmembrane adaptor protein, is initially described as a receptor-activating subunit component of natural killer cells[75,76]. TYROBP is expressed in a variety of lymphocytes, peripheral blood monocytes, macrophages and dendritic cells[77]. TYROBP is also related to activation of immunoreceptors, which contribute to several biological functions[78]. This was consistent with the results of immune infiltration analysis.

The full name of FPR1 is neutrophil formyl peptide receptors 1, which generates signals to mediate neutrophil activation during inflammation[79,80]. Recently, the role of FPR1 in a range of respiratory conditions has been intensively studied. For example, FPR1 expression is increased in peripheral neutrophils of patients with chronic obstructive pulmonary disease and emphysema[81,82]. Blockade of FPR1 reduces many essential functions of neutrophils associated with defense, such as superoxide anion generation, elastase release, chemotaxis and phagocytosis[83].

FCGR3B gene belongs to the FCGR family. A meta-analysis of human genome-wide association study reveals that Fc$\gamma$R (FCGR) gene is located at an important asthma susceptibility locus[84]. Fc$\gamma$Rs are glycoproteins that bind the Fc region of immunoglobulin G. Fc$\gamma$Rs mediate a variety of immune functions such as antigen presentation, immune complex clearance, pathogen phagocytosis, degranulations, and cytokine production[85].

We further inferred the upstream signaling pathways of key targets in asthma, and the results showed that PPAR, IL-1 and MAPK + PI3K were of great significance for exploring the asthma severity mechanisms. Janette K Burgess et al. confirm that IL-1$\beta$ contribute to regulating airway inflammation and remodeling[86]. Resistin-like molecule -$\beta$ may promote AR in asthma through the ERK/MAPK-PI3K/Akt signaling pathway[87]. As multifunctional molecules, PPARs are implicated in a variety of a human diseases such as cancer[88,89], metabolic[90] and autoimmune conditions[91].

To further illustrate the relevance of the 46 targets screened from the key modules related to asthma severity, we constructed PPI and GGI networks for interaction analysis. The sources of the interaction relationship between the various proteins in the PPI network included text mining, experiments, databases, co-expression, neighborhood-joining, gene fusion and cooccurrence, and the interactions between every two proteins was scored. We ultimately obtained 10 key targets including *PTPRC, TLR2, MMP9, FCGR3B, TYROBP, CXCR1, S100A12, FPR1, CCR1* and *CXCR2*. GeneMANIA further demonstrated the interaction between these 10 targets. The results showed that the top 10 hub genes interacted with *S100A8, S100A9, LILRB3, FCER1G, MCER1G* and *MMP25*. Functional enrichment analysis showed that these genes were mainly related to myeloid leukocyte migration, immune receptor activity, leukocyte chemotaxis, neuroinflammatory response, neutrophil migration, regulation of

inflammatory and cell chemotaxis, which further demonstrated the importance of these 10 targets in the immune regulation of asthma severity. Meanwhile, Spearman correlation analysis between 10 key genes and 28 immune cells illustrated that all genes exhibited associations with multiple immune cells, especially neutrophils.

Synergistic effect between immune cells and genes involved in immune regulation is essential for immune response. With this in mind, associations between 10 biomarkers related to asthma severity and 29 immune cells were analyzed to gain a deeper insight into the immune mechanisms underlying asthma progression. Richard Y Kim et al. confirm that IL-1$\beta$ is related to neutrophil airway inflammation, disease severity and steroid resistance, and thus can be considered as a possible target for SA treatment[92]. Stephanie A Christenson also discusses the therapeutic potential of the NLRP3 inflammasome/caspase-1/IL-1$\beta$ axis in severe steroid-resistant asthma[93]. CCR1 antagonists can reduce eosinophil differentiation and interfere with the development of allergic airway inflammation[94]. Ting Zhou et al. reveal that soluble CD14 shed from CD14 can clinically regulate the activation and function of T lymphocytes and is negatively associated with asthma severity[95].

Neutrophils are the first line of defense against foreign invaders. The effective execution of many neutrophil effector responses requires the presence of $\beta_2$ integrins[96]. The protein encoded by FCGR3B is a low-affinity receptor for the Fc region of gamma immunoglobulin and is a common immune marker for systemic lupus erythematosus[97]. Furthermore, we constructed a neutrophil-dominant asthma mouse model, which showed elevated *PTPRC, TLR2, MMP9, FCGR3B, TYROBP, CXCR1, FPR1, CCR1* and *CXCR2* mRNA expression. Finally, RT-qPCR analysis confirmed the elevated expression levels of 9 key targets in addition to *PTPRC* in an in vitro model of LPS-induced bronchial epithelial cell injury.

The CTD database was used to predict the underlying therapeutic agents associated with asthma severity, and we noted that Reparixin may have great significance for precision treatment of asthma, targeting *MMP9, CXCR1* and *CXCR2*.

These small molecule compounds or drugs may have some potential in the treatment of asthma, but further clinical or basic research is needed to confirm these results. Cytochalasin D and Methotrexate may play vital roles in asthma. Cytochalasin D eliminates the uptake of apoptotic eosinophils by airway epithelial cells[98]. Although not yet proven, Methotrexate has been shown that effective treatment of allergic inflammation may prevent long-term consequences of asthma and avert deterioration of pulmonary function[99]. Reparixin is a noncompetitive CXCR1/2 antagonist[100]. The CXCLs/CXCR2 axis mediates the arrival of neutrophils at inflammatory sites, and plays an important role in anti-infection and control of pathogen invasion[101]. Thus, Reparixin may target CXCLs-CXCR1/2 as a therapeutic strategy. To verify the efficacy of Reparixin, we designed an LPS-induced cell model for in vitro validation, and found that Reparixin exhibited a protective effect on LPS-injured 16HBE cells. Then, RT-qPCR analysis confirmed that Reparixin can notably reduce the expression levels of *CXCR1, CXCR2*, and *MMP9*.

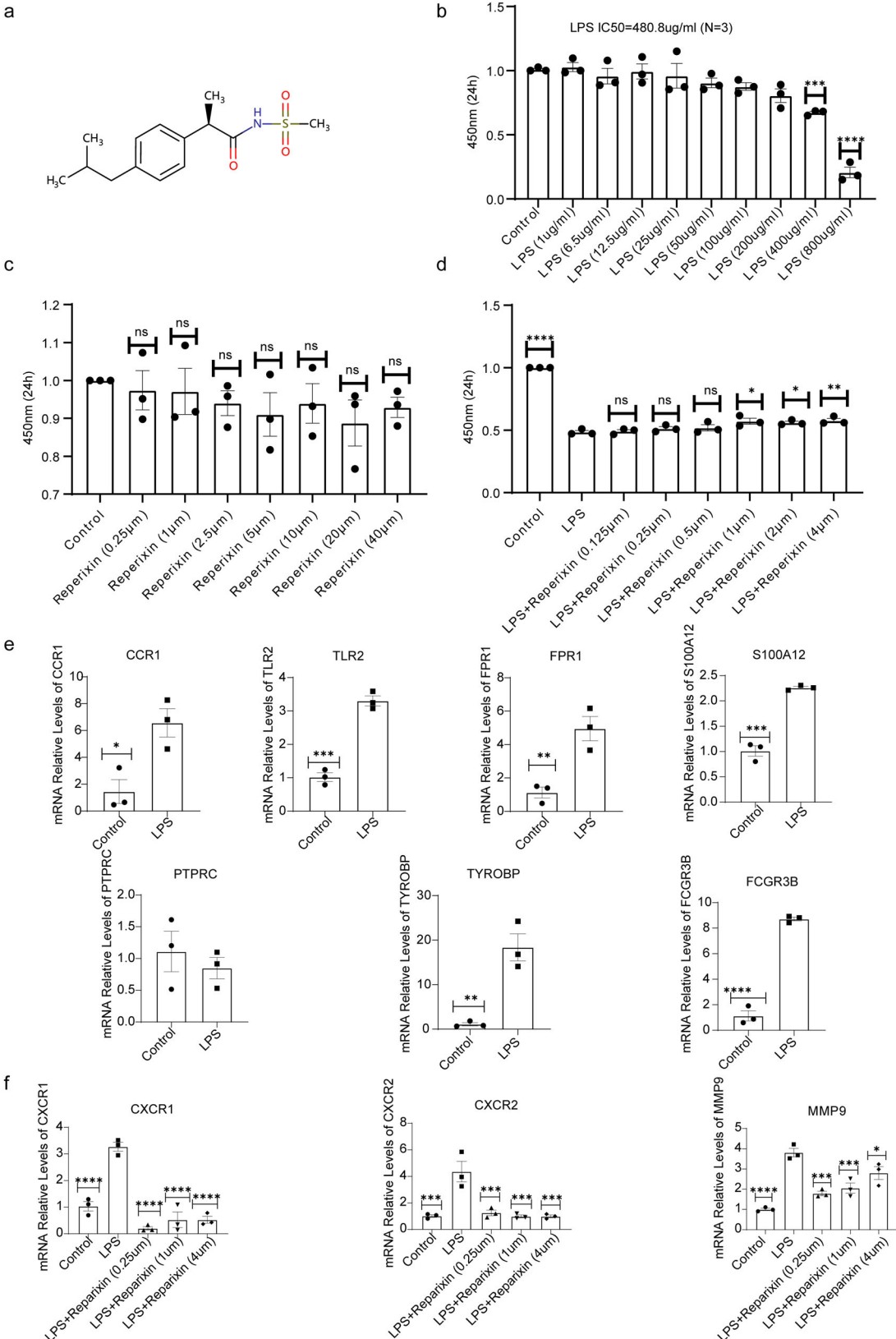

**Fig. 10 Reperixin antagonized LPS-induced cell injury in the human bronchial epithelial cell line 16HBE. a** The chemical structure of Reperixin; **b** The viability of 16HBE cells induced by different concentrations of LPS (from 1 ug/mL to 800 ug/mL) was detected by CCK-8 assay (N = 3); **c** 16HBE cells were incubated with various concentrations of Reperixin (from $0\,\mu M$ to $40\,\mu M$) for 24 h, and then cell viability was analyzed by CCK8 assay (N = 3); **d** Cells were treated with the indicated does of Reperixin ($1\,\mu M$, $2\,\mu M$ and $4\,\mu M$) prior to 480.8 $\mu g/mL$ LPS exposure for 24 h, and then cell viability was evaluated by CCK8 method (N = 3). **e** RT-qPCR analysis of hub genes in control and LPS-induced 16HBE cells (N = 3). **f** Reperixin suppressed the expression of CXCR1, CXCR2 and MMP9 in LPS-injured 16HBE cells (N = 3). *$p < 0.05$; **$p < 0.01$; ***$p < 0.001$; ****$p < 0.0001$. Error bars indicate mean ± SEM.

## Conclusion

This study deciphered the hub immune genes and the pattern of immune infiltration related to the asthma severity. Asthma severity was correlated with the level of neutrophil infiltration, and FEV1 and FVC were negatively correlated with neutrophil infiltration. WGCNA was then utilized to identify the genes most closely associated with neutrophils and asthma severity in the turquoise module. *PTPRC*, *TLR2*, *MMP9*, *FCGR3B*, *TYROBP*, *CXCR1*, *S100A12*, *FPR1*, *CCR1* and *CXCR2* were considered as the key genes related to asthma severity. Compared to healthy, mild and moderate asthma, immature mast cells, neutrophils, plasmacytoid dendritic cells, plasmacytoid dendritic cells and regulatory T cells were more abundant in SA samples, which may be associated with the expression of 10 hub genes. Reparixin will be promising in the treatment of asthma. Finally, we constructed NA mouse model and LPS-stimulated human 16HBE cell model to successfully verify the 10 key targets expression. In summary, this study suggested that using airway epithelial cell transcriptomics to understand the immune infiltration mechanism related to asthma severity is of great clinical value. Thus, our findings may guide future validation of clinical samples.

## Limitation

Lung function, a key indicator for evaluating asthma progression, was not tested due to limitations in sampling at the end of the experiment. In addition, our analysis and validation of the findings were limited to clinical, animal and cellular data. The clinical samples were obtained from the GEO database without additional large-scale validation in clinical settings.

## Study approval

This study was approved by the Laboratory Animal Ethics Committee of Guangzhou University of Chinese Medicine (Approval No. 20230413001). We have complied with all relevant ethical regulations for animal use.

## Data availability

The numerical source data underlying all graphs can be found in Supplementary Data 1. The data underlying this article is available at the Gene Expression Omnibus public repository (GSE89809).

## Code availability

The R code underlying this article is available at 4TU.RearchData (https://doi.org/10.4121/19537258.v1).

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

## Acknowledgements

We thank the Lingnan Medical Research Center of Guangzhou University of Chinese Medicine and the Famous Traditional Chinese Medicine inheritance physician unit of Xiao-Hong Liu of Guangdong for their support. We thank Professor LingJun Wang for his support in carrying out this work. We thank Professor Xiaohong Liu for her kind help in the initiation of this research and her selfless assistance. This work was supported by National Natural Science Foundation of China (Grant No. 82204985) and National Natural Science Foundation of Guangdong, China (Grant no. 2023A1515010807 and 2021A1515010146). This study was supported by the "Excellent doctoral dissertation Cultivation Fund of the First Affiliated Hospital of Guangzhou University of Chinese Medicine" (YB202302). This study was supported by the Guangzhou Science and Technology Bureau, Guangzhou, China (Grant NO.202102080453). This work was supported by the Project of Department of Education of Guangdong Province (Grant no. 2022KQNCX013) and Science and Technology Project of GuangZhou City (Grant No.2023A4J1854). This study was also supported by administration of Traditional Chinese Medicine of Guangdong Province (Grant no. 20221144). This study was supported by the Sanming Project of Medicine in Shenzhen (Grant no. SZZYSM202106006 and SZZYSM202206013). This work was also supported by the "Double First-Class" and High-level University Discipline Collaborative Innovation Team Project of Guangzhou University of Chinese Medicine (Grant No.2021XK16), the Key-Area Research and Development Program of Guangdong Province (Grant No. 2020B1111100002), and the Technology Research of COVID-19 Treatment and Prevention and Special Project of Traditional Chinese Medicine Application-Research on the platform construction for the prevention and treatment of viral infectious diseases with traditional Chinese medicine (Grant No. 2020KJCX-KTYJ-130).

## Author contributions

XF.H, Y.J. XH. L and SF.Z. participated in the guidance and revision of the entire article. Q.Y. conducted the writing, main experiments and data analysis, and XX.Z. Y.X. and J.Y. performed the RT-qPCR experiment. XX.Z., MF.Z., CX.L, XY.L. and WJ.Z. performed the literature searches and data analysis. Y.X. and XX.Z conducted the revision of the article during the revision phage.

## Competing interests

The authors declare no competing interests.
