## [Peer Review File · Communications Biology]

Reviewers' comments:

Reviewer #1 (Remarks to the Author):

Asthma is now considered a heterogeneous inflammatory disease of the airways that has 4 defined phenotypes based on the types of infiltrating or resident immune cells and their related regulatory factors. Of the subtypes of asthma, type 2 or eosinophilic, has received the most experimental and clinical attention: this focus has led to the development of biological agents that target key regulatory check points and subsequent enhanced therapeutic benefit.

Much less is known about the pathognomonic mechanisms of the remaining 3 subtypes of asthma. Of these, type 1 asthma or neutrophilic, is very predominant (-30- 50%) and these patients are often refractive to gold standard steroid and bronchodilator therapy leading to uncontrolled exacerbations that can be life-threatening or require hospitalisation.

In the manuscript by Qian Yan et al., state-of-the-art genomic and bioinformatic techniques are employed to interrogate the transcriptome in non-type 2 asthma of varying severity. A major finding was significant differences in the degree of neutrophil infiltration in asthma with the degree of severity: increase neutrophils with increased severity. For example, the degree of neutrophil infiltration was shown to be negatively related to FEV1% and FEV1/FVC. This finding in-itself is not new or unexpected.

There was an enrichment in cell chemotaxis and cytokine-cytokine receptor interactions in the turquoise module. Further SPEED2 indicated that IL-1, TNF- α and MAPK+PI3K may be important upstream effector molecules. Again, this is not new knowledge but important confirmatory data of previous studies, obtained by employing state-of-the-art approaches.

The Spearman correlation analysis showed that 10 significant genes were linked to various immune cells. This work extends current knowledge. Furthermore, 10 key genes identified in asthma were verified in a OVA/CFA45 induced mice model of asthma. RT-qPCR showed the high expression of AQP9, CCR1, CD14, FCER1G, FGR, ITGB2, CXCR4 and IL-1B, which were related to the asthma severity and neutrophils. Using a mouse model to look for expression of candidate genes, except for FCGR3B and CCL18, 8 of the critical genes, including IL-1B, ITGB2, CCR1, FGR, CD14, CXCR4, FCER1G and AQP9 were highly expressed during active disease.

This identified potential molecules that can be used for drug development. Inhibition of some of these candidate molecules as proof of principle would have significantly advanced the paper.

The paper is overall well written, logical and well controlled. The concept of repurposing specific drugs provides a next step based on potential previous clinical trials for alternative conditions. An excellent study.

Reviewer #2 (Remarks to the Author):

General comment:

The authors have investigated immune infiltration mechanisms, molecular biomarkers and pathways related to asthma severity using transcriptomics analysis. They have chosen a published gene expression dataset, which are from airway epithelium of mild, moderate and severe asthma patients. The bioinformatic approaches used allowed them to establish gene network and to identify key genes. In this regard, the authors have made a plausible effort in improving current asthma therapy for asthma and potentially for other chronic lung diseases. However, there are a few weaknesses that may damage the merit for publication in its current form. These issues are listed as below.

- 1.The authors purposely selected airway epithelium. However, there are many immune cells with high ssGSEA score. The author should verify this with real biological samples, e.g. airway brushing, and compare those to the level of airway epithelial cells. It is surprise that the authors did not analysis eosinophils. Eosinophils had higher ssGSEA score than neutrophils and are one of most important immune cells for asthma.
- 2.Airway epithelial cells are at the front line of the body and importantly regulate subsequent immune responses. Why is there no gene expression signature of epithelial cells? It is odd that the authors selected dataset of airway epithelium but there was no result of epithelial cells.
- 3.The health control should not be excluded. Proper health control is essential for the study. If there are cross-platform dataset used, the authors should avoid cross-platform bias.
- 4.To verify gene network and to identify hub genes, TFs, or pathways, the authors at least need to perform in vitro experiments to confirm their findings.
- 5.The authors attempted to identify potential drugs. What are the target genes/TFs/Pathways of those drugs? Again, the author should at least perform in vitro experiments to confirm their findings.
- 6.The establishment of asthma model very likely results in extremely high eosinophilic inflammation. This model is therefore contradictory to the results of authors' analysis, with a focus on neutrophils. In addition, the animal model should be fully characterized, e.g. AHR, airway inflammation/BALF, mucus production, cytokine profile in lung/airway etc. HE score is a very crude measurement. It is very hard to tell what occurs in the histological images.
- 7.This manuscript should be edited professionally. E.g. The title is confusing. The discussion needs revision.
- 8.There were some letters missing. E.g. in Figure 2A.
- 9.What do all those GSMs stand for? i.e. Figure 3A.

Response to the Review Comments

Date: 2023-10-15

Journal: COMMUNICATIONS BIOLOGY

Manuscript Number: COMMSBIO-23-1304A

Title: *Bronchial Epithelial Transcriptomics and experimental validation reveal the asthma severity-related neutrophilic signatures and potential treatments*

Authors: Qian Yan, Xinxin Zhang, Jing Yang, Chengxin Liu, Miaofen Zhang, Wenjiang Zheng, Xueying Lin, Hui-ting Huang, Xiaohong Liu, Yong Jiang, Shao-feng Zhan, Xiufang Huang

Thank you for the reviewers' comments concerning our manuscript entitled "**Bronchial Epithelial Transcriptomics and experimental validation reveal the asthma severity-related neutrophilic signatures and potential treatments**". We would like to thank you and the reviewers again for taking the time to review our manuscript. In this revision, we made significant revisions to our research in response to the questions raised by you and the reviewers. We not only improved the approach of data analysis and further optimized the OVA/CFA-induced neutrophil asthma model, but also constructed an LPS-induced bronchial epithelial cell injury model. In addition, we validated potential drugs for key targets. Here are our main revision directions:

(1) During data analysis, we realized that the source of our samples had not been described clearly enough in previous articles. Therefore, we clearly described that our samples were primarily epithelial cells collected by brushing the airways of real-world asthma patients (originated from the GEO database). Therefore, we have corrected the statements in the original manuscript regarding the source of clinical bronchial epithelial cell samples.

(2) We incorporated the gene expression characteristics of epithelial cells into our initial analysis, including the control group with no cross-platform variations.

(3) Both *in vivo* and *in vitro* experiments were performed in order to validate the identified key targets; in addition, *in vitro* experiments and human bronchial epithelial

cell line were used to assess the potential effects of the predicted drugs on the predicted target genes.

(4) By introducing supplementary PAS and Masson staining techniques to evaluate the mucus production and airway remodeling, and quantifying immune cells in mouse BAL fluid, along with ELISA detection of neutrophil-related cytokines (e.g., IL-1 β , IL-8, TNF- α), and MPO measurements, we improved the OVA/CFA-induced NA mouse model.

(5) We have completely revised the entire text, paying particular attention to the title and discussion sections. To ensure clarity, we have highlighted all revisions in blue. In addition, we had the manuscript meticulously revised by a professional English speaker.

Finally, we would like to clarify that due to the active participation of the author, Jing Yang, in the animal experiments and cellular studies during the revision process, Jing Yang was designated as the co-first author. All authors are aware of and agree with this decision. We thank the editors and reviewers for their valuable comments. Please feel free to contact me if there are any further questions or if you need additional information. Your patience and consideration are greatly appreciated.

Sincerely,

Xiufang Huang

huangxiufang@gzzyydx17.wecom.work

Reviewers' comments:

Reviewer #1 (Remarks to the Author):

Asthma is now considered a heterogeneous inflammatory disease of the airways that has 4 defined phenotypes based on the types of infiltrating or resident immune cells and their related regulatory factors. Of the subtypes of asthma, type 2 or eosinophilic, has received the most experimental and clinical attention: this focus has led to the development of biological agents that target key regulatory check points and subsequent enhanced therapeutic benefit.

Much less is known about the pathognomonic mechanisms of the remaining 3 subtypes of asthma. Of these, type 1 asthma or neutrophilic, is very predominant (-30- 50%) and these patients are often refractive to gold standard steroid and bronchodilator therapy leading to uncontrolled exacerbations that can be life-threatening or require hospitalisation.

In the manuscript by Qian Yan et al., state-of-the-art genomic and bioinformatic techniques are employed to interrogate the transcriptome in non-type 2 asthma of varying severity. A major finding was significant differences in the degree of neutrophil infiltration in asthma with the degree of severity: increase neutrophils with increased severity. For example, the degree of neutrophil infiltration was shown to be negatively related to FEV1% and FEV1/FVC. This finding in-itself is not new or unexpected.

There was an enrichment in cell chemotaxis and cytokine-cytokine receptor interactions in the turquoise module. Further SPEED2 indicated that IL-1, TNF- α and MAPK+PI3K may be important up stream effector molecules. Again, this is not new knowledge but important confirmatory data of previous studies, obtained by employing state-of-the-art approaches.

The Spearman correlation analysis showed that 10 significant genes were linked to various immune cells. This work extends current knowledge. Furthermore, 10 key genes identified in asthma were verified in a OVA/CFA induced mice model of asthma. RT-qPCR showed the high expression of AQP9, CCR1, CD14, FCER1G, FGR, ITGB2, CXCR4 and IL-1B, which were related to the asthma severity and neutrophils. Using a mouse model to look for expression of candidate genes, except for FCGR3B and CCL18,

8 of the critical genes, including IL-1B, ITGB2, CCR1, FGR, CD14, CXCR4, FCER1G and AQP9 were highly expressed during active disease.

This identified potential molecules that can be used for drug development. Inhibition of some of these candidate molecules as proof of principle would have significantly advanced the paper.

The paper is overall well written, logical and well controlled. The concept of repurposing specific drugs provides a next step based on potential previous clinical trials for alternative conditions. An excellent study.

Reply: Thank you for acknowledging the improvements made in our article. In this revision, we have not only enhanced the data analysis section but also introduced additional methodologies, including PAS and Masson staining, ELISA detection of IL-8/MPO and TNF- α , as well as cell classification and quantification in mouse BALF. Building upon your suggestion, we further validated the expression of key targets in 16HBE cells and verified the effects of potential drugs. The *in vitro* results showed that the 9 key targets were significantly elevated (except for PTPRC), and at the same time, Reperixin could act as an inhibitor of IL-8, targeting and inhibiting the expression of MMP9, CXCR1 and CXCR2. Thus, we not only confirmed the 10 key targets identified by the *in vivo* experiments at the cellular level, but also observed changes in the expression of MMP9, CXCR1, and CXCR2 molecules after administration of Reperixin. We would like to thank you for your support and acknowledgement. Below are the results of the *in vitro* cell experiments:

Fig 11. Reperixin antagonized LPS-induced cell injury in the human bronchial epithelial cell line 16HBE. (A) The chemical structure of Reperixin; (B) The viability of 16HBE cells induced by different concentrations of LPS (from 1 $\mu\text{g}/\text{mL}$ to 800 $\mu\text{g}/\text{mL}$) was detected by CCK-8 assay (N=3); (C) 16HBE cells were incubated with various concentrations of Reperixin (from 0.25 μM to 40 μM) for 24 h, and then cell viability

was analyzed by CCK8 assay (N=3); (D) Cells were treated with the indicated doses of Reperixin (1 μ M, 2 μ M and 4 μ M) prior to 480.8 μ g/mL LPS exposure for 24 h, and then cell viability was evaluated by CCK8 method. (E-K) RT-qPCR analysis of hub genes in control and LPS-induced 16HBE cells (N=3). (L-N) Reperixin suppressed the expression of CXCR1, CXCR2 and MMP9 in LPS-injured 16HBE cells (N=3). *. $p < 0.05$; **: $p < 0.01$; ***: $p < 0.001$; ****: $p < 0.0001$.

Reviewer #2 (Remarks to the Author):

General comment:

The authors have investigated immune infiltration mechanisms, molecular biomarkers and pathways related to asthma severity using transcriptomics analysis. They have chosen a published gene expression dataset, which are from airway epithelium of mild, moderate and severe asthma patients. The bioinformatic approaches used allowed them to establish gene network and to identify key genes. In this regard, the authors have made a plausible effort in improving current asthma therapy for asthma and potentially for other chronic lung diseases. However, there are a few weaknesses that may damage the merit for publication in its current form. These issues are listed as below.

Reply: Thank you for your review of our paper. We appreciate your feedback and are glad that you find our approach plausible in improving asthma therapy and potentially other chronic lung diseases. We also appreciate that you have identified some weaknesses in the current form of the paper. We will consider your comments and address these issues accordingly to strengthen the merit of the paper for publication. Thank you again for your time and effort in reviewing our paper.

1. (1) The authors purposely selected airway epithelium. However, there are many immune cells with high ssGSEA score. The author should verify this with real biological samples, e.g., airway brushing, and compare those to the level of airway epithelial cells.

Reply: Thank you for your comments on our manuscript. We appreciate your valuable comments and suggestions to improve our work. We apologize for the confounding of the data analyzed in this study. The samples used in this study were actually from “epithelial cell brushes” from asthma patients of varying severity, (<https://www.ncbi.nlm.nih.gov/geo/query/acc.cgi?acc=GSE89809>), and we mistakenly referred to these data as “airway epithelial cells”. Thank you for pointing this out.

We carefully considered your suggestion, and the Xcell method was applied for quantifying the 64 types of immune and stromal cells, including epithelial cells, eosinophils and neutrophils. We calculated the enrichment scores for each cell type

signature using the ssGSEA algorithm before converting them into cell type scores using a specially designed algorithm. We found that the infiltration levels of epithelial cells and neutrophils were higher in severe asthma, which is consistent with our findings. We also compared the infiltration levels of neutrophils and epithelial cells among different groups, and found that the score of epithelial cells was higher than that of neutrophils, but the difference between the two was not statistically significant. We apologize again for our mistake and thank you for pointing out our issue.

Fig. The Xcell method used to evaluate the infiltration of neutrophils and bronchial epithelial cells between healthy controls, mild, moderate and severe asthma patients.

(2) It is surprise that the authors did not analysis eosinophils. Eosinophils had higher ssGSEA score than neutrophils and are one of most important immune cells for asthma.

Reply: Thank you for your comment. As you pointed out, we did not analyze the potential significance of eosinophils in different severities of asthma. While we quantified the differences in eosinophil and neutrophil infiltration between healthy individuals, mild asthmatics, moderate asthmatics, and severe asthmatics using multiple methods, our results showed that while eosinophil infiltration levels were higher than neutrophil infiltration levels, no statistically significant difference was observed among the four groups. It is worth noting that increased levels of neutrophils have previously been associated with severe asthma and asthma exacerbation (PMID:

9309987;10556116;10712320). In our study, we found that the level of neutrophil infiltration increased with asthma progression, with statistically significant differences observed between severe and mild asthma. Therefore, we focused on exploring the potential significance of neutrophil infiltration differences in asthma progression. Finally, we validated our findings through the construction of both a neutrophil asthma mouse model and a bronchial epithelial cell injury model. We appreciate your valuable comments and thank you again for taking the time and effort to review our work.

Fig. The methods of CIBERSORT, ssGSEA, ConsensusTME, and Xcell used to evaluate the infiltration of neutrophils and eosinophils between healthy controls, mild, moderate and severe asthma patients.

2. *Airway epithelial cells are at the front line of the body and importantly regulate subsequent immune responses. Why is there no gene expression signature of epithelial cells? It is odd that the authors selected dataset of airway epithelium but*

there was no result of epithelial cells.

Reply: Thank you for your comments on our manuscript. We appreciate your valuable feedback and suggestions for improving our work. Regarding your concern about the lack of gene expression signature of epithelial cells in our study, we would like to clarify that although we did not perform differential analysis specifically for gene expression signature of epithelial cells, we have used WGCNA to identify modules and key genes related to asthma severity and neutrophil infiltration. The study by Bodie-Curren et al. indicates that IL-33 blockade improves the severity of asthma exacerbation by attenuating neutrophil recruitment and downstream generation of NETs (PMID: 37506849). We believe that this method can better find the module genes related to neutrophils and severe asthma.

However, we understand your suggestion and agree that it would be beneficial to describe the gene expression profile of airway epithelial cells in more detail. Therefore, we included additional analyses of epithelial cell gene expression in our revised manuscript to provide a more comprehensive understanding of our findings. We appreciate your helpful comments, and we will carefully consider all of your suggestions as we revise our manuscript. Thank you once again for your time and effort in reviewing our work.

“To elucidate the molecular mechanisms underlying asthma progression, we conducted DEGs and functional enrichment analyses on gene expression profiles in asthma of different severities. Based on the above screening criteria, we analyzed DEGs in microarray data from healthy controls, mild, moderate and SA patients. Comparing mild asthma patients to healthy controls, we identified a total of 76 up-regulated and 37 down-regulated DEGs (Fig. 3A). Enrichment analysis revealed that these DEGs primarily participated in biological processes (BPs) such as “O-glycan processing” and “negative regulation of endopeptidase activity” (Fig. 3B). Remarkably, the “IL-17 signaling pathway” emerged as the only significant signaling pathway, with MUC5AC, MUC5B and CXCL8 exhibiting significant enrichment.

Between moderate and mild asthma, 42 up-regulated and 103 down-regulated DEGs were identified (Fig. 3C). Enrichment analysis revealed that “negative regulation

of substrate adhesion-dependent cell spreading” was the most significant BP (Fig. 3D). KEGG analysis showed that the “HIF-1 signaling pathway” was the most important signaling pathway, with TFRC, NOS2 and ERBB2 enriched in this pathway. ERBB2 (HER2), the second member of the EGFR family[29], is also involved in epithelial repair in asthmatic patients.

Between severe and moderate asthma, 293 up-regulated and 289 down-regulated DEGs were identified (Fig. 3E). BPs analysis showed that “antigen processing and presentation of endogenous antigen” were the most important BPs (Fig. 3F). KEGG analysis showed that “Cell adhesion molecules”, “Longevity regulating pathway”, and “Cellular senescence” were significantly enriched.

Based on the aforementioned results, we explored in greater depth the molecular features that influence asthma progression. We identified 259 up-regulated and 604 down-regulated DEGs between SA and mild asthma patients (Fig. 3G). BPs analysis indicated the highest enrichment of “neutrophil degranulation” and “neutrophil activation involved in immune response”, which also included processes such as “neutrophil chemotaxis” and “neutrophil migration” (Fig. 3H). Furthermore, KEGG enrichment analysis highlighted the significant enrichment of pathways such as “Osteoclast differentiation” and “Viral protein interaction with cytokine and cytokine receptor”. Consequently, a total of 863 DEGs were detected between SA and mild asthma for further analysis”.

Fig 3. Differential gene analysis between healthy controls, mild, moderate, and severe asthma. (A-B). the volcano plot between the mild asthma and healthy control groups, along with the enrichment analysis of biological processes. (C-D), (E-F), and (G-H). the volcano plots between moderate and mild asthma patients, severe and moderate asthma patients, respectively, alongside the corresponding enrichment analysis of biological processes.

3. *The health control should not be excluded. Proper health control is essential for the study. If there are cross-platform dataset used, the authors should avoid cross-platform bias.*

Reply: Thank you for your review and feedback. We appreciate you bringing up the issue of excluding the healthy control group in our study and acknowledge that it was an oversight on our part. We understand the importance of including the healthy control group in our analysis, and we have since incorporated this healthy control into our study and further confirmed our results. Thank you again for your valuable comments and attention to detail, which will help us improve our research.

Fig 2A. Scatter map of the degree of 28 immune cells; NES: Normalized Enrichment Scorew.

4. *To verify gene network and to identify hub genes, TFs, or pathways, the authors at least need to perform in vitro experiments to confirm their findings.*

5. *The authors attempted to identify potential drugs. What are the target genes/TFs/Pathways of those drugs? Again, the author should at least perform in vitro experiments to confirm their findings.*

Reply to questions 4-5: Thank you for your appreciation of our manuscript and your valuable review comments. During the revision process, we established a model of LPS-induced bronchial epithelial cell injury and confirmed the expression levels of 10 key targets and potential drugs in an in vitro cell model. RT-qPCR analysis confirmed that the expression of 10 key targets were also upregulated in LPS-damaged 16HBE cells (except for PTPRC). Importantly, Reparixin was able to reduce the expression

levels of CXCR1, CXCR2 and MMP9, and the figure is shown below:

Fig 11. Reperixin antagonized LPS-induced cell injury in the human bronchial epithelial cell line 16HBE. (A) The chemical structure of Reperixin; (B) The viability of 16HBE cells induced by different concentrations of LPS(from 1 ug/mL to 800 ug/mL) was detected by CCK-8 assay (N=3); (C) 16HBE cells were incubated with various

concentrations of Reperixin (from 0.25 μ M to 40 μ M) for 24 h, and then cell viability was analyzed by CCK8 assay (N=3); (D) Cells were treated with the indicated doses of Reperixin (1 μ M, 2 μ M and 4 μ M) prior to 480.8 μ g/mL LPS exposure for 24 h, and then cell viability was evaluated by CCK8 method. (E-K) RT-qPCR analysis of hub genes in control and LPS-induced 16HBE cells (N=3). (L-N) Reperixin suppressed the expression of CXCR1, CXCR2 and MMP9 in LPS-injured 16HBE cells (N=3). *. $p < 0.05$; **. $p < 0.01$; ***. $p < 0.001$; ****. $p < 0.0001$.

6. *The establishment of asthma model very likely results in extremely high eosinophilic inflammation. This model is therefore contradictory to the results of authors' analysis, with a focus on neutrophils. In addition, the animal model should be fully characterized, e. g. AHR, airway inflammation/BALF, mucus production, cytokine profile in lung/airway etc. HE score is a very crude measurement. It is very hard to tell what occurs in the histological images.*

Reply: Thank you for your appreciation of our manuscript and your valuable review comments. We are grateful for your feedback regarding the establishment of the asthma model and the necessity of its comprehensive characterization. We acknowledge that in our previous animal model, we conducted HE staining and serum IgE ELISA analysis, which may have led to confusion as it closely resembles the eosinophilic asthma model. In light of your suggestions, we have focused our efforts on analyzing and characterizing a neutrophil-dominated model during the revision process. By doing so, we aim to provide a more accurate representation and understanding of the specific aspects and implications of this particular model. We genuinely appreciate your insightful comments and guidance in improving the clarity and accuracy of our research. Therefore, in this revision process, we mainly analyzed the neutrophil-dominated model.

The establishment of the mouse neutrophil asthma model (**Figure 10A**) in our study primarily relied on the significant increase in neutrophil count within the model group. To assess the pathological conditions of the mouse lung tissue, we conducted a thorough examination. Histopathological analyses, such as HE staining, proved to be effective in indicating the presence of inflammatory cell infiltration (**Figure 10B**). Notably, the model group exhibited pronounced inflammatory cell infiltration around the airways, with significant differences observed in the HE scores. We also employed

Periodic acid-Schiff (PAS) staining to demonstrate increased mucus secretion and increased goblet cell proliferation in the neutrophil asthma model (**Figure 10C**). Furthermore, Masson's trichrome staining revealed elevated collagen deposition in the neutrophil model compared to the control group (**Figure 10D**). Collectively, these histopathological findings suggest a certain degree of lung tissue injury following the administration of OVA/CFA. These comprehensive histopathological assessments provide robust evidence supporting the existence of lung tissue damage in the mouse model induced by OVA/CFA.

Subsequent analysis of the bronchoalveolar lavage (BAL) fluid cell counts further substantiated our findings. It revealed a significant increase in the total number of leukocytes in the model group, with a particular emphasis on the elevated number of neutrophils (**Figure 10E**). This observation reaffirms the presence of heightened neutrophilic inflammation in the mouse model, confirming its suitability as a neutrophil asthma model. The increased neutrophil count underscores the relevance and effectiveness of this model for studying neutrophil-driven asthma pathogenesis.

Then, we conducted an analysis of various neutrophil-related factors in both serum and BAL fluid, including myeloperoxidase (MPO), IL-8, TNF- α , and IL-1 β . Neutrophils actively contribute to inflammation and airway remodeling by producing chemokines and proteases, which are key components in neutrophilic asthma [PMID: 28784414]. MPO, being predominantly produced by neutrophils, serves as a specific marker for neutrophil activation [PMID: 10619791]. We observed high MPO expression in the model group, indicating successful construction of the neutrophil asthma model. In neutrophilic asthma, dysfunctions of the airway epithelium and secretion of several inflammatory mediators by airway epithelial cells have been documented. These mediators include IL-1 β , TNF- α , and the well-known neutrophil chemoattractant, IL-8 [PMID:28163052; 32526308]. Increased expression of IL-1 β has been consistently associated with neutrophilic asthma [PMID: 28252317]. Neutrophilic airway inflammation, disease severity, and steroid resistance have been linked to IL-1 β expression in human asthma. Moreover, reducing neutrophil counts inhibits IL-1 β -induced steroid-resistant airway hyperresponsiveness [PMID: 28252317]. Studies have

shown that the expression of TNF- α plays an important role in the alteration of inflammatory environment in neutrophilic asthma [PMID: 34291112]. Furthermore, the CXC chemokine IL-8 has been shown to facilitate neutrophil migration through stimulation of CXCR1 and CXCR2 receptors [PMID: 32800946]. Patients with neutrophilic asthma have exhibited higher levels of IL-8 and TNF- α in sputum samples when compared to non-neutrophilic asthma [PMID: 30105264] (**Figure 10F**). By examining these neutrophil-related factors, we were able to further validate the presence and characteristics of neutrophilic inflammation in our model. These findings contribute to a better understanding of the underlying mechanisms and potential therapeutic targets in neutrophilic asthma.

Fig. 10. Construction of an OVA/CFA-induced asthma mouse model. (A). Flowchart of mouse model establishment; (B-D). HE, PAS and Masson's trichrome staining of lungs from control and model groups. Scale bars were 100 μ M and 200 μ M. The respective pathology scores were summarized in the right panel (N=3). (E). Wright-Giemsa staining of bronchoalveolar lavage (BAL) fluid. Scale bars were 100 μ M and 50 μ M. The graph on the right showed the total cell counts and four different cell counts (neutrophils, eosinophils, lymphocytes and macrophages) in the BAL, respectively (N=3), and the blue arrow represented neutrophils. (F). Total IgE level in serum (N=3),

MPO (N=5), IL-1 β (N=3) and IL-8 (N=5) levels in serum and BAL, and TNF- α (N=5) level in serum were measured by ELISA; (G). RT-qPCR analysis of hub genes by 2^{- $\Delta\Delta$ Ct} method (N=5). *. $p < 0.05$; **. $p < 0.01$; ***. $p < 0.001$; ****. $p < 0.0001$.

At present, the results we have presented provide sufficient evidence to support the successful construction of a mouse model of neutrophilic asthma. Moreover, it is consistent with previous research that has utilized complete Freund's adjuvant (CFA) as an adjuvant to induce a neutrophil-dominated asthma model. For instance, Mengling Xia et al. conducted a study investigating whether neutrophil activation and NETosis play a significant role in airway inflammation in neutrophilic asthma using OVA/CFA/LPS-induced mouse models [PMID: 36271366]. Similarly, Hui-Hsien Pan et al. developed an OVA/CFA sensitization model to induce neutrophil-dominant asthma, demonstrating elevated levels of airway hyperresponsiveness (AHR), IgE, infiltration of inflammatory cells (particularly neutrophils) in the lungs, and overall airway inflammation [PMID: 34649239]. Given the consistent findings reported in previous studies, we chose to utilize the OVA/CFA-induced model for the induction of neutrophilic asthma based on the established evidence and methodology available. This approach strengthens the validity and relevance of our model, allowing for meaningful comparisons and further understanding of neutrophilic asthma.

However, it is crucial to acknowledge that airway hyperresponsiveness (AHR) is a critical indicator of lung function in asthmatic individuals. Due to the limitations in sample acquisition conditions during the study, we were unable to conduct AHR detection within the appropriate timeframe. Consequently, this represents a limitation of our study. However, we have included it in the analysis of limitations section in our manuscript. By acknowledging this constraint, we aim to provide a comprehensive and transparent evaluation of our study's findings and ensure the proper interpretation of the results.

“Our study has some limitations. ”

“Lung function, a key indicator for evaluating asthma progression, was not tested due to limitations in sampling at the end of the experiment. Since our analysis and validation of the findings were limited to clinical, animal and cellular data. The clinical samples were obtained from the GEO database without additional large-scale

validation in clinical settings” (Page 22-23, Line680-684).

We gratefully appreciate for your valuable suggestions which will make our research more complete and persuasive. Thanks again for the professional advice.

7. *This manuscript should be edited professionally. E.g., The title is confusing. The discussion needs revision.*

Reply: Thank you for your review of our manuscript and for bringing up the issue with the title and discussion section. We appreciate your suggestions that the manuscript should be edited professionally. We acknowledge that there may be some issues with the clarity of the title and the discussion section, and we will take this feedback into consideration when making revisions to the manuscript. We recognize the importance of presenting our research in a clear and concise manner to ensure that readers can easily understand the purpose and implications of our findings. In addition, we have modified the discussion section and highlighted the modified section in blue font. Thank you again for your constructive comments and for helping us improve our manuscript, and the new title was as follows:

“Bronchial Epithelial Transcriptomics and experimental validation reveal the asthma severity-related neutrophilic signatures and potential treatments”.

8. *There were some letters missing. E.g., in Figure 2A.*

Reply: Thank you for your review of our manuscript and for bringing up the issue with missing letters in Figure 2A. We apologize for any confusion caused by this error and have taken steps to correct it in this revisions of the manuscript. We appreciate your attention to detail and your helpful comments, which will help us to improve the quality of our research and presentation of our findings. Thank you again for your valuable feedback.

9. *What do all those GSMs stand for? i.e. Figure 3A.*

Reply: Thank you for your review and comment on our Figure 3A. We apologize for any confusion caused by the use of GSMs in the figure. GSM stands for “Gene Expression Sample Matrix” and is a unique identifier used in gene expression analysis to distinguish between different samples. In Figure 3A, the GSMs represent the different airway epithelium samples from mild, moderate, and severe asthma patients

that were used in our transcriptomics analysis. We appreciate your feedback and will make sure to provide more clarity in the figure legends and text in future revisions of the paper. Thank you again for your helpful comments.

REVIEWERS' COMMENTS:

Reviewer #1 (Remarks to the Author):

Improved version

Reviewer #2 (Remarks to the Author):

The authors have satisfactorily replied my comments.